# FROM GENERALIST TO SPECIALIST: ADAPTING VISION LANGUAGE MODELS VIA TASK-SPECIFIC VISUAL INSTRUCTION TUNING

## ABSTRACT

Large vision language models (VLMs) combine large language models with vision encoders, demonstrating promise across various tasks. However, they often underperform in task-specific applications due to domain gaps between pre-training and fine-tuning. We introduce VITask, a novel framework that enhances task-specific adaptability of VLMs by integrating task-specific models (TSMs). VITask employs three key strategies: exemplar prompting (EP), response distribution alignment (RDA), and contrastive response tuning (CRT) to improve the task-specific performance of VLMs by adjusting their response distributions. EP allows TSM features to guide VLMs, while RDA enables VLMs to adapt without TSMs during inference by learning from exemplar-prompted models. CRT further optimizes the ranking of correct image-response pairs, thereby reducing the risk of generating undesired responses. Experiments on 12 medical diagnosis datasets across 9 imaging modalities show that VITask outperforms both vanilla instruction-tuned VLMs and TSMs, showcasing its ability to integrate complementary features from both models effectively. Additionally, VITask offers practical advantages such as flexible TSM integration and robustness to incomplete instructions, making it a versatile and efficient solution for task-specific VLM tuning.

## 1 INTRODUCTION

Large Vision Language Models (VLMs) combine the capabilities of large language models (LLMs) with pre-trained vision encoders, enabling them to process and understand both text and images Liu et al. (2023a; 2024b); Driess et al. (2023); gpt; Dai et al. (2024); Chen et al. (2023b; 2024); Alayrac et al. (2022); Bai et al. (2023). This integration allows VLMs to perceive visual inputs, comprehend complex queries, and perform sophisticated reasoning across a wide array of tasks and domains. The success of VLMs drives the growing trend of adapting VLMs for a wide range of task-specific applications such as medical diagnosis, autonomous driving, and content creation He et al. (2024); Moor et al. (2023); Li et al. (2024b); Wu et al. (2023); Zhou et al. (2024a); Xu et al. (2024).

Despite the wide applicability of VLMs, recent studies have noted that their performance often often falls short compared to task-specific models (TSMs) when fine-tuned for specific tasks or domains Singhal et al. (2023); Yang et al. (2024). The performance gap between VLMs and TSMs represents a critical limitation, particularly in real-world scenarios that demand high accuracy and reliable service quality. Although substantial progress has been made in enhancing the performance and versatility of VLMs Wu et al. (2023); Liu et al. (2023b); Lai et al. (2024); Wang et al. (2024), most of these approaches do not focus on effectively adapting *pre-trained* VLMs to *specific* tasks or datasets. This leads to a fundamental question: *can we adapt VLMs to perform as well as, or even surpass, task-specific models?*

In this study, we use image classification as a case study to investigate why fine-tuned VLMs often lag behind TSMs in performance. We identify two main factors contributing to this decline: 1) Un-specialized Image Representations: Image features learned during pre-training for vision-language tasks are not effective for specific classification tasks. They often miss important details needed for these tasks, making it hard for the vision encoder to extract useful information. 2) Indirect Tuning

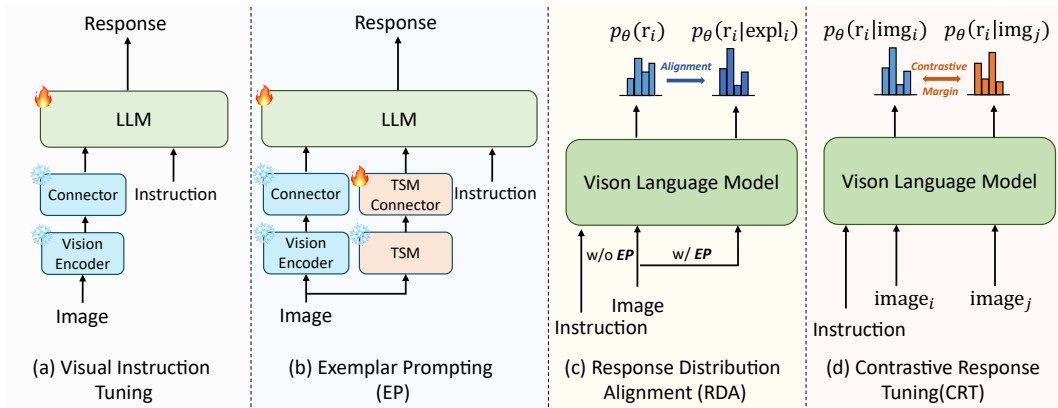

Figure 1: Overview of the proposed VITask framework. (a) Traditional visual instruction tuning. (b) Exemplar Prompting (EP) enhances VLM's image representations using TSM features without modifying pre-trained features. (c) Response Distribution Alignment (RDA) aligns EP and non-EP responses to capture task-specific information. (d) Contrastive Response Tuning (CRT) leverages negative samples to improve the VLM's response ranking capability by maximizing the margin between correct and incorrect image-response pairs.

Objective: Fine-tuning VLMs typically emphasizes enhancing text generation, such as predicting the next word, rather than directly addressing image classification. This approach can hinder the models from learning the essential features required for effective image classification, resulting in subpar performance.

To address these challenges, we propose VITask, a novel framework that combines the strengths of TSMs and VLMs to improve task-specific performance without sacrificing the versatility and instruction-following capabilities of VLMs. Our main idea leverages small, easily obtainable TSMs and a task-specific tuning objective to improve the learning of desired response distributions. To maintain the vision-language alignment in pre-trained VLMs, we avoid directly updating the vision encoder for new tasks. Instead, we propose *exemplar prompting*, using TSM features as *exemplars* to enhance VLM adaptability without altering pre-trained image features, while incorporating specialized task representations. Additionally, we introduce response distribution alignment to align the response distributions between VLMs with and without exemplar prompting. This allows the VLM to implicitly learn from the TSM by utilizing its own responses during fine-tuning. Finally, we propose *contrastive response tuning*, which maximizes the likelihood of correct image-response pairs (e.g., $p(\texttt{cat}|\texttt{<cat image>})$) while minimizing the likelihood of incorrect pairs (e.g., $p(\texttt{cat}|\texttt{<dog image>})$). This approach promotes more discriminative and accurate response rankings for visual instructions, thereby enhancing task-specific performance.

We evaluate VITask on 12 medical image diagnosis datasets and show that it consistently outperforms both TSMs and vanilla instruction-tuned VLMs. Furthermore, VITask demonstrates robustness to incomplete instructions, providing flexibility for real-world applications where task descriptions may not be comprehensive. Our results highlight the potential of VITask to generalize beyond medical tasks, making it a versatile framework for task-specific VLM tuning.

## 2 RELATED WORK

**Large Vision Language Models.** Vision Language Models (VLMs) are multimodal models designed to process and understand both visual and textual information. Inspired by the success of large language models (LLMs), such as GPT-4 Achiam et al. (2023), LLaMA-2 Touvron et al. (2023), and PaLM-2 Anil et al. (2023), the development of VLMs has evolved from simply aligning image-text pairs, as seen in models like CLIP Radford et al. (2021), BLIP Li et al. (2022), to integrating vision encoders into LLMs, enabling them to process and interpret visual information. Examples of such models include GPT-4V [1], InstructBLIP Dai et al. (2024), PaLM-E Driess et al. (2023), MiniGPT-4 Zhu et al. (2024), LLaVA series Liu et al. (2023a; 2024a;b), InternVL Chen et al. (2023b; 2024), the Gemini series Team et al. (2023); Reid et al. (2024), Claude-3 Anthropic (2024),

and Qwen-VL-Max Bai et al. (2023). Recent advancements in VLMs focus on improving model architectures Liu et al. (2024a); Chen et al. (2023b; 2024), training strategies Liu et al. (2024d;e); He et al. (2023), and datasets Yu et al. (2023); Li et al. (2023b); Liu et al. (2024c); Li et al. (2023a), resulting in enhanced capabilities and broader applications.

**Visual Instruction Tuning.** Current VLM training pipelines usually follows a two-stage protocol. First, the vision language alignment stage align the image features from the vision encoder with the word embeddings encoded in LLMs. Second, the visual instruction tuning stage adapts VLMs to follow instructions that involve both visual and textual inputs, making VLMs able to respond to natural language commands or questions based on the content of an image Liu et al. (2023a); Dai et al. (2024). Visual instruction tuning is a crucial step for making VLMs more interactive, versatile, and context-aware, allowing them to follow instructions related to specific tasks, enhancing its accuracy and adaptability to real-world applications where users provide visual and textual inputs. There are many existing works in the field of visual instruction tuning. Typical research topics focus on gaining specialized visual understanding ability Yue et al. (2024); Nisar et al. (2024); Chen et al. (2023a); Lai et al. (2024), reducing computational costs Hu et al. (2021); Luo et al. (2024); Lee et al. (2024), mitigating hallucination Leng et al. (2024); Zhou et al. (2024b); Hu et al. (2023), creating or augmenting instruction data Yu et al. (2023); Li et al. (2023b); Liu et al. (2024c); Li et al. (2023a).

**Integrating VLMs and TSMs.** Several approaches have been proposed to integrate VLMs with task-specific models in an attempt to leverage the strengths of both Liu et al. (2023b); Lai et al. (2024); Li et al. (2024a). However, these works primarily focus on utilizing TSMs as task-specific heads or tools for constructing a new VLM, without addressing the challenges of fine-tuning *pre-trained* VLMs for specific tasks or datasets. Our work focuses on improving the visual instruction tuning paradigm to achieve better task-specific performance, especially when the model faces domain gaps with downstream task data.

# 3 INSTRUCTION-TUNED VLMS VS. TASK-SPECIFIC MODELS

In this section, we compare instruction-tuned VLMs with TSMs to evaluate their performance on domain-specific tasks. While instruction-tuned VLMs are designed to handle both image and text inputs in a generalized manner, TSMs are optimized for a particular task or dataset, often leading to superior performance for specific applications. Despite the wide range of potential downstream tasks, image classification serves as a fundamental task for benchmarking. We thus conduct a head-to-head comparison between the VLMs and TSMs on a single classification task, as a case study for our analysis.

**Setting.** We consider fine-tuning a pre-trained VLM and a naïve task-specific model on a given classification dataset, which may have domain gaps with the data used for pre-training. Specifically, we use InternVL2-2B Chen et al. (2024) as the pre-trained VLM and a ViT-Base model Dosovitskiy et al. (2020) pre-trained on ImageNet-21k Deng et al. (2009), with a randomly initialized linear classification head, as the task-specific classifier. Both models are fine-tuned for multi-class image classification on the HAM10000 dataset Tschandl et al. (2018), which contains 10,015 dermatoscopic images across 7 classes for diagnosing pigmented skin lesions. We follow the same setting in Yang et al. (2023) to set the training, validation, and test set as 70%, 10%, 20%, respectively. In what follows, we conduct our analysis within this setting for simplicity and validate our findings through formal experiments on 12 medical datasets across 6 domains, as detailed in Section 5.

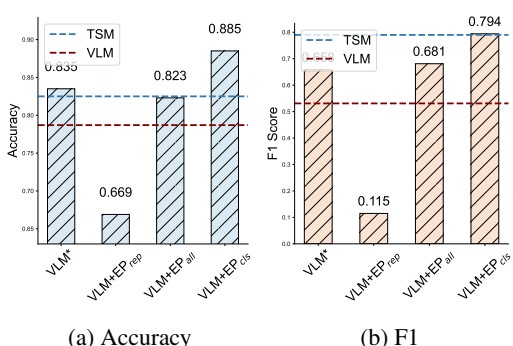

(a) Accuracy  (b) F1

Figure 2: Illustration of the performance discrepancy between TSM and VLMs.

**Instruction Formatting.** Since the classification dataset is not originally designed for instruction tuning, we convert the training data into an instruction-following format as follows He et al. (2024):

```
<|user|><image>{instruction}<|assistant|>{response}
```

Here, the tags `<|user|>` and `<|assistant|>` are used to indicate instruction-following for ease of reading and do not affect the experimental results. The `<image>` tag represents the image features extracted from the vision encoder of the pre-trained VLM. Using this format, an instruction for the HAM10000 dataset Tschandl et al. (2018) could be: "*Analyze the given dermatoscope image for diagnosis. The possible diagnoses are:* {*possible disease names*}.". The corresponding response for an image with vascular lesions would be `vascular lesions`.

**Model Training.** For VLMs, we follow the common practice of instruction-tuning the LLM component while keeping the vision encoder and vision-language connector frozen, utilizing LoRA Hu et al. (2021) to improve training efficiency. For TSMs, we fully fine-tune the ViT classifier using class labels, updating both the ViT model and the classification head during training. More implementation details are provided in Section 5.

**Observations.** As shown in Figure 2, the ViT classifier (TSM) achieves an F1 score of 0.790, significantly outperforming the instruction-tuned VLM (VLM for short subsequently), which only reaches an F1 score of 0.531. This highlights the difficulty of fine-tuning VLMs for specific tasks. The large performance gap likely stems from the fact that pre-trained image features may not encompass all the essential representations required for new tasks. When the VLM's vision encoder is made trainable (denoted by VLM*), the model's performance improves to an F1 score of 0.658, which, while better than VLM, still lags behind TSM. It is worth noting that although making the vision encoder trainable enhances performance, this approach may be undesirable, as it risks distorting the valuable vision-language alignment and conversational abilities that VLMs rely on. These findings suggest that vanilla visual instruction tuning may struggle when adapted to specific downstream tasks, facing unique challenges in achieving task-specific performance on par with TSMs. This is particularly notable given that TSMs are generally much smaller and easier to train for specialized tasks. *Can we adapt a VLM to achieve comparable or superior task-specific performance while preserving its pre-trained vision-language alignment and conversational abilities?*

## 4 TASK-SPECIFIC VISUAL INSTRUCTION TUNING

In this section, we investigate why fine-tuned VLMs may underperform in classification tasks and highlight two key issues in the current visual instruction tuning paradigm: 1. Unspecialized Image Representations: The pre-trained vision encoder learns representations optimized for vision-language alignment, which are often sub-optimal for downstream classification tasks. 2. Indiect Tuning Objective: The tuning objective focuses on next token prediction, which is more suited to text generation than to classification tasks that require fine-grained discrimination. To overcome these challenges, we proposed VITask, a novel framework (Figure 1) that bridges TSMs and VLMs to enhance task-specific adaptability and performance.

**Exemplar Prompting.** We first introduce Exemplar Prompting (EP). A VLM takes a visual image $\mathbf{v}$ and a textual instruction $\mathbf{x}$ as inputs, aiming to generate relevant and helpful response $\mathbf{y}$. Visual instruction tuning can be framed as conditional probability estimation $p_\theta(\mathbf{y} \mid \mathbf{v}, \mathbf{x})$, where $\theta$ represents the learnable parameters of the VLM. Given a visual instruction dataset $D = \{\text{image}_i, \text{instruction}_i, \text{response}_i\}_{i=1}^N$ containing $N$ image-instruction-response triples, visual instruction tuning adapts the VLM by minimizing the following objective:

$$\mathcal{L}^{\text{Van}} = \frac{1}{N} \sum_{i=1}^N -\log p_\theta(\text{response}_i \mid \text{image}_i, \text{instruction}_i). \tag{1}$$

For image classification, we can train a TSM, such as the ViT classifier mentioned in Section 3, on the same dataset $D$ without instruction formatting and extract the latent feature for each $\text{image}_i$. We define this latent feature as $\text{exemplar}_i$ for $\text{image}_i$. Exemplar prompting utilizes the TSM features to

*prompt* VLMs during fine-tuning by augmenting the VLM's image features $\text{image}_i$ with $\text{exemplar}_i$. This is achieved by modifying the tuning objective (1) as follows:

$$\mathcal{L}^{\text{EP}} = \frac{1}{N} \sum_{i=1}^{N} - \log p_\theta(\text{response}_i \mid \text{image}_i, \text{exemplar}_i, \text{instruction}_i). \tag{2}$$

The rationale behind exemplar prompting is that since the TSM is optimized to learn specialized features for downstream tasks, it can offer task-specific latent features that guide the VLM in learning a better mapping between the visual instruction and the desired response. This enhances the VLM's adaptability without directly altering its pre-trained image features, thereby preserving the vision-language alignment while incorporating relevant task-specific knowledge.

**Implementation and Analysis.** As shown in Figure 1, we implement exemplar prompting by introducing a learnable vision-language connector to align TSM features with the LLM of the VLM. This connector is updated along with the LLM, while the vision encoders of both VLM and TSM remain frozen during fine-tuning. For a ViT classifier as the TSM, exemplars can be derived from all patch embeddings ($\text{EP}_{\text{all}}$), the CLS token ($\text{EP}_{\text{cls}}$), or by replacing all VLM image features with TSM features ($\text{EP}_{\text{rep}}$). From Figure 2, we observe that replacing all VLM image features with TSM features results in poor performance, showing that TSM features alone cannot maintain VLMs' instruction-following ability for new tasks. However, exemplar prompting with all patch embeddings or the CLS token significantly boosts classification performance compared to standard instruction tuning. Notably, VLM+$\text{EP}_{\text{cls}}$ matches or even exceeds the performance of both TSM and VLM with a trainable vision encoder, demonstrating that incorporating just one TSM feature (CLS token) enhances task-specific instruction-response mappings. Conversely, using all patch tokens ($\text{EP}_{\text{all}}$) is less effective, suggesting that irrelevant features may degrade performance. Therefore, if not specified otherwise, we use the CLS token for EP, considering it is the most effective and efficient.

> **Takeaway #1:** TSM features can prompts VLMs to generate desired responses.

**Response Distribution Alignment.** One key intuition behind exemplar prompting is that it creates a shortcut between exemplars and desired responses, making instruction-following easier. While effective, using exemplars requires combining TSM and VLM during both fine-tuning and inference. This increases the size of the model, which may be impractical when dealing with multiple tasks and corresponding TSMs. A natural question arises: can task-specific adaptability be improved without relying on TSMs and exemplars during inference? The answer is yes. Instead of explicitly learning the exemplar-response mapping, we propose Response Distribution Alignment (RDA) to implicitly learn the distribution of desired responses. The idea is for the VLM with exemplar prompting to "teach" the VLM without exemplar prompting during fine-tuning. Specifically, we

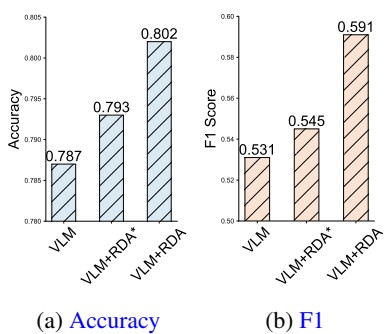

(a) Accuracy      (b) F1

Figure 3: Illustration of RDA effectiveness.

minimize the Kullback-Leibler (KL) divergence between the response distributions of VLM and VLM+EP:

$$\mathcal{L}^{\text{RDA}} = \frac{1}{N} \sum_{i=1}^{N} D_{\text{KL}}(p_\theta(\text{response}_i) \| p_\theta(\text{response}_i \mid \text{exemplar}_i)), \tag{3}$$

where we omit the common conditions on $\text{image}_i$ and $\text{instruction}_i$ in the response distributions for simplicity. This approach allows the VLM to learn specialized task information from TSM by mimicking the behavior of VLM+EP, all without requiring exemplars during inference.

**Implementation and Analysis.** The proposed RDA strategy optimizes (3) alongside the basic objectives in (1) and (2). Since our aim is to learn from the exemplar-prompted VLM rather than the other way around, we detach the gradient of the exemplar-prompted distribution $p_\theta(\text{response}_i \mid$

exemplar$_i$) when computing (3). Figure 3 demonstrates the impact of RDA on classification performance. We also test a variant, RDA$^*$, which is identical to RDA but without gradient detachment. The results show that VLM+RDA improves the F1 score by $6\%$, demonstrating that TSM can effectively guide VLM to learn a better response distribution even without using exemplar prompting during inference. In contrast, VLM+RDA$^*$ shows no significant improvement over the baseline VLM, verifying that RDA's gains are due to the task-specific information transferred from VLM+EP.

> **Takeaway #2:** VLMs can implicitly acquire task-specific knowledge from TSM.

**Contrastive Response Tuning.** The success of response distribution alignment suggests that we do not need to teach VLM explicit mappings from instructions to responses; instead, these mappings can be implicitly learned by refining the distribution of desired responses. Motivated by Hewitt et al. (2024), we propose the concept of *visual response ranking capability*, referring to a VLM's ability to assign a higher likelihood to correct image-response pairs than to incorrect ones for a given instruction. For two independent image-instruction-response triples $(\text{image}_i, \text{instruction}_i, \text{response}_i)$ and $(\text{image}_j, \text{instruction}_j, \text{response}_j)$, with $\text{instruction}_i = \text{instruction}_j$ and $\text{response}_i \neq \text{response}_j$, the visual response ranking capability holds for a VLM $p_\theta$ if

$$p_\theta(\text{response}_i \mid \text{image}_i, \text{instruction}_i) > p_\theta(\text{response}_i \mid \text{image}_j, \text{instruction}_i), \qquad (4)$$

where we assume the instruction $\text{instruction}_i$ is the same for both triples for clarity. Intuitively, a VLM with this capability will more likely generate correct responses for visual instructions. The degree to which a VLM possesses this ranking capability reflects how well it can differentiate between correct and incorrect image-response pairs for a given instruction. We argue that *vanilla visual instruction tuning often fails to establish this ranking capability* because it focuses solely on learning instruction-response mappings and does not explicitly account for the critical relationship between images and responses. As a result, an instruction-tuned VLM might rank incorrect image-response pairs higher than the correct ones, leading to suboptimal performance on specific tasks. To address this issue, we propose Contrastive Response Tuning (CRT) to maximize the margin between correct and incorrect image-response pairs. This is done by minimizing the following objective:

$$\mathcal{L}^{\text{CRT}} = \frac{1}{N} \sum_{i=1}^{N} - \log q_\theta(\text{response}_i \mid \text{image}_i, \text{image}_j, \text{instruction}_i), \qquad (5)$$

where the *margin distribution* is defined as:

$$q_\theta(\text{response}_i \mid \text{image}_i, \text{image}_j, \text{instruction}_i) = \text{Softmax}(\mathbf{y}_i^{\text{pos}} - \mathbf{y}_i^{\text{neg}}). \qquad (6)$$

Here, $\mathbf{y}_i^{\text{pos}}$ represents the logits for the *positive* response distribution $p_\theta(\text{response}_i \mid \text{image}_i, \text{instruction}_i)$, and $\mathbf{y}_i^{\text{neg}}$ represents the logits for the *negative* response distribution $p_\theta(\text{response}_i \mid \text{image}_j, \text{instruction}_i)$. CRT encourages the model to maximize the likelihood of the correct image-response pair (positive) while minimizing the likelihood of incorrect pairs (negative), thus promoting more discriminative and accurate response rankings. This approach enhances the VLM's visual response ranking capability, improving task-specific adaptability and accuracy in scenarios like image classification.

**Implementation and Analysis.** For each triple $(\text{image}_i, \text{instruction}_i, \text{response}_i) \sim D$, we randomly select a negative $\text{image}_j$ from another triple $(\text{image}_j, \text{instruction}_j, \text{response}_j) \sim D$, ensuring that $\text{instruction}_i = \text{instruction}_j$ and $\text{response}_i \neq \text{response}_j$. Then, CRT (5) can be applied to each token of $\text{response}_i$ given $\text{image}_i$, $\text{image}_j$, and $\text{instruction}_i$ autoregressively. To gain a deeper understanding of how CRT improves the visual response ranking capability, we evaluate its effect on the HAM1000 test set. We compute the average probability of each token in $\text{response}_i$ for both positive and negative image pairs based on three different VLMs: a pre-trained VLM without fine-tuning, a VLM tuned with vanilla visual instruction tuning, and a VLM tuned with our CRT strategy. Figure 4 illustrates the normalized density of response probabilities for positive and negative image pairs across these VLMs. Figure 4a shows that the pre-trained VLM, without any fine-tuning, does not possess the visual response ranking capability, as the probability distributions for positive and negative image pairs are nearly identical. This confirms that the pre-trained VLM lacks task-specific instruction-following ability. Figure 4b indicates that while vanilla instruction tuning enables the VLM to some extent to differentiate between positive and negative image pairs,

there remains a significant overlap. Many incorrect image-response pairs still receive high probabilities, posing a risk of undesired responses. Figure 4c demonstrates that CRT effectively sharpens the distinction between correct and incorrect image-response pairs by maximizing the margin distribution $q_\theta(\text{response}_i \mid \text{image}_i, \text{image}_j, \text{instruction}_i)$. The CRT-tuned VLM shows a clear increase in the probability for correct image-response pairs and a corresponding decrease for incorrect ones, signifying that CRT substantially enhances the model's ability to generate desirable and accurate responses compared to vanilla instruction-tuned VLMs.

> **Takeaway #3:** Contrastive response tuning improves the visual response ranking capability.

**VITask Framework.** To bring together all the proposed strategies, we introduce the VITask framework, a two-stage pipeline designed for task-specific visual instruction tuning, analogous to the way VLMs are trained. Stage 1: we make the task-specific connector learnable and fine-tune the VLM using vanilla visual instruction tuning in conjunction with EP and RDA. The objective for this stage is: $\mathcal{L}^{\text{Stage1}} = \mathcal{L}^{\text{Van}} + \mathcal{L}^{\text{EP}} + \alpha \mathcal{L}^{\text{RDA}}$. The primary goal of Stage 1 is to establish the basic visual instruction-following ability and learn an effective task-specific connector that aligns TSM features with the LLM. Stage 2: After the task-specific connector is trained, we freeze it and then fine-tune the VLM with all the proposed loss functions. The objective becomes:

$$\mathcal{L}^{\text{Stage2}} = \mathcal{L}^{\text{Van}} + \mathcal{L}^{\text{EP}} + \alpha \mathcal{L}^{\text{RDA}} + \beta \mathcal{L}^{\text{CRT}}, \tag{7}$$

where $\alpha$ and $\beta$ adjust the weight of $\mathcal{L}^{\text{RDA}}$ and $\mathcal{L}^{\text{CRT}}$, respectively. In this stage, the model fine-tunes its visual response ranking capability through CRT while maintaining the learned visual-instruction mapping from Stage 1. Although so far our framework and analysis focus on a single task and dataset, VITask can be generalized to multi-task or multi-dataset settings by expanding the label space and training a joint TSM. This flexibility allows the framework to build more robust, domain-specific VLMs, capable of handling a variety of downstream tasks.

**Advantages.** VITask offers several advantages beyond improving task-specific performance. One major benefit is its ability to *decouple image representation learning from visual instruction tuning* by incorporating TSMs into VLMs. This flexibility allows for the use of any TSM architecture, giving practitioners the freedom to choose the best model for their specific task. Furthermore, once fine-tuned, the VLM can perform inference without needing the TSM, maintaining task-specific adaptability while reducing model complexity.

Another key advantage of VITask is its *plug-and-play collaboration* between VLMs and TSMs. When a new task is introduced, a new TSM can be separately trained and directly connected to the VLM without requiring further instruction tuning. Since TSMs are generally smaller and easier to train than VLMs, VITask provides an efficient way to adapt VLMs to new tasks, making the framework highly scalable and adaptable to multiple domains.

Additionally, VITask demonstrates *robustness against the content of instructions*. Instruction-tuned VLMs often rely on carefully crafted instructions for optimal performance. For instance, in experiments with the HAM10000 dataset, detailed class information is typically included in the instruction to enhance accuracy. However, in real-world applications, users may not always know such detailed information in advance. VITask mitigates this limitation by adapting the response distribution based on task-specific information from TSMs rather than solely relying on the instruction itself, enabling strong performance even with more generalized or incomplete instructions.

## 5 EXPERIMENTS

In this section, we evaluate the proposed VITask framework in fine-tuning a VLM for medical diagnosis. Our experimental setup is designed to test the following key aspects: 1) the ability of VITask to improve task-specific classification performance; 2) the flexibility of VITask in adapting to various tasks without retraining the entire model; 3) the robustness of VITask against incomplete instructions.

**Datasets and Metrics.** We utilize the MedMNIST 2D Dataset collection Yang et al. (2023) for fine-tuning and testing our VLM. This comprehensive collection encompasses 9 distinct biomedical

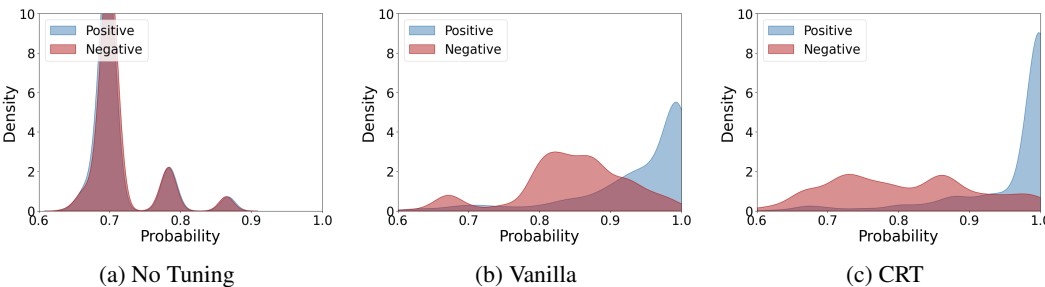

(a) No Tuning      (b) Vanilla      (c) CRT

Figure 4: Illustration on how CRT improves the visual response ranking capability for VLMs.

Table 1: Performance of VLMs on medical image diagnosis tasks. and * denotes results from the original paper He et al. (2024).

| Dataset | Metric | TSM | MedDr* | Qwen2 VL 7B | LLaVA 13B | LLaVA Med | +VITask w/o EP | +VITask w/ EP | InternVL 2B | +VITask w/o EP | +VITask w/ EP |
|---|---|---|---|---|---|---|---|---|---|---|---|
| PathMNIST | Accuracy↑ | 0.933 | - | 0.823 | 0.935 | 0.939 | $0.940^{+0.1\%}$ | $0.964^{+2.5\%}$ | 0.926 | $0.939^{+1.3\%}$ | $0.953^{+2.7\%}$ |
| | Macro-F1↑ | 0.926 | - | 0.754 | 0.905 | 0.915 | $0.916^{+0.1\%}$ | $0.949^{+3.4\%}$ | 0.896 | $0.911^{+1.5\%}$ | $0.937^{+4.1\%}$ |
| ChestMNIST | Accuracy↑ | 0.533 | 0.519 | 0.510 | 0.535 | 0.513 | 0.510 | $0.518^{+0.5\%}$ | 0.523 | 0.513 | 0.517 |
| | Macro-F1↑ | 0.095 | 0.134 | 0.051 | 0.073 | 0.088 | $0.107^{+1.9\%}$ | $0.118^{+3.0\%}$ | 0.024 | $0.102^{+7.8\%}$ | $0.129^{+10.5\%}$ |
| DermaMNIST | Accuracy↑ | 0.846 | 0.690 | 0.716 | 0.731 | 0.800 | $0.832^{+3.2\%}$ | $0.856^{+5.6\%}$ | 0.770 | $0.810^{+4.0\%}$ | $0.877^{+10.7\%}$ |
| | Macro-F1↑ | 0.792 | 0.395 | 0.384 | 0.355 | 0.556 | $0.672^{+11.6\%}$ | $0.723^{+16.7\%}$ | 0.499 | $0.633^{+13.4\%}$ | $0.772^{+27.3\%}$ |
| OCTMNIST | Accuracy↑ | 0.934 | 0.692 | 0.738 | 0.788 | 0.868 | $0.870^{+0.2\%}$ | $0.942^{+7.4\%}$ | 0.726 | $0.853^{+12.7\%}$ | $0.952^{+22.6\%}$ |
| | Macro-F1↑ | 0.941 | 0.661 | 0.729 | 0.786 | 0.868 | $0.869^{+0.1\%}$ | $0.942^{+7.4\%}$ | 0.704 | $0.846^{+14.2\%}$ | $0.952^{+24.8\%}$ |
| Pneumonia-MNIST | Accuracy↑ | 0.968 | 0.929 | 0.438 | 0.881 | 0.910 | $0.918^{+0.8\%}$ | $0.952^{+4.2\%}$ | 0.886 | $0.888^{+0.2\%}$ | $0.931^{+4.5\%}$ |
| | Macro-F1↑ | 0.965 | 0.926 | 0.383 | 0.864 | 0.900 | $0.909^{+0.9\%}$ | $0.923^{+2.3\%}$ | 0.873 | 0.872 | $0.923^{+5.0\%}$ |
| RetinaMNIST | Accuracy↑ | 0.472 | - | 0.280 | 0.557 | 0.542 | $0.650^{+10.8\%}$ | $0.650^{+10.8\%}$ | 0.590 | $0.625^{+3.5\%}$ | $0.632^{+4.2\%}$ |
| | Macro-F1↑ | 0.424 | - | 0.166 | 0.279 | 0.280 | $0.466^{+18.6\%}$ | $0.544^{+26.4\%}$ | 0.370 | $0.457^{+8.7\%}$ | $0.522^{+15.2\%}$ |
| BreastMNIST | Accuracy↑ | 0.897 | 0.878 | 0.494 | 0.750 | 0.212 | $0.821^{+60.9\%}$ | $0.859^{+64.7\%}$ | 0.744 | $0.846^{+10.2\%}$ | $0.865^{+12.1\%}$ |
| | Macro-F1↑ | 0.866 | 0.842 | 0.510 | 0.671 | 0.382 | $0.802^{+42.0\%}$ | $0.833^{+45.1\%}$ | 0.524 | $0.798^{+27.4\%}$ | $0.828^{+30.4\%}$ |
| BloodMNIST | Accuracy↑ | 0.987 | 0.955 | 0.286 | 0.951 | 0.975 | $0.977^{+0.2\%}$ | $0.987^{+1.2\%}$ | 0.931 | $0.983^{+5.2\%}$ | $0.991^{+6.0\%}$ |
| | Macro-F1↑ | 0.990 | 0.954 | 0.166 | 0.832 | 0.856 | $0.860^{+0.4\%}$ | $0.867^{+1.1\%}$ | 0.818 | $0.864^{+4.6\%}$ | $0.870^{+5.2\%}$ |
| TissueMNIST | Accuracy↑ | 0.697 | - | 0.575 | 0.613 | 0.642 | $0.665^{+2.3\%}$ | $0.755^{+11.3\%}$ | 0.569 | $0.643^{+7.4\%}$ | $0.761^{+19.2\%}$ |
| | Macro-F1↑ | 0.681 | - | 0.411 | 0.497 | 0.540 | $0.569^{+2.9\%}$ | $0.685^{+14.5\%}$ | 0.419 | $0.538^{+11.9\%}$ | $0.690^{+27.1\%}$ |
| OrganAMNIST | Accuracy↑ | 0.934 | 0.846 | 0.807 | 0.878 | 0.916 | $0.934^{+1.8\%}$ | $0.953^{+3.7\%}$ | 0.828 | $0.924^{+9.6\%}$ | $0.955^{+12.7\%}$ |
| | Macro-F1↑ | 0.950 | 0.822 | 0.777 | 0.855 | 0.908 | $0.927^{+1.9\%}$ | $0.947^{+3.9\%}$ | 0.801 | $0.917^{+11.6\%}$ | $0.950^{+14.9\%}$ |
| OrganCMNIST | Accuracy↑ | 0.869 | - | 0.724 | 0.796 | 0.865 | $0.893^{+2.8\%}$ | $0.922^{+5.7\%}$ | 0.778 | $0.889^{+11.1\%}$ | $0.920^{+14.2\%}$ |
| | Macro-F1↑ | 0.898 | - | 0.681 | 0.750 | 0.843 | $0.875^{+3.2\%}$ | $0.909^{+6.6\%}$ | 0.742 | $0.871^{+12.9\%}$ | $0.908^{+16.6\%}$ |
| OrganSMNIST | Accuracy↑ | 0.726 | - | 0.672 | 0.689 | 0.738 | $0.769^{+3.1\%}$ | $0.799^{+6.1\%}$ | 0.635 | $0.758^{+12.3\%}$ | $0.809^{+17.4\%}$ |
| | Macro-F1↑ | 0.737 | - | 0.618 | 0.621 | 0.687 | $0.719^{+3.2\%}$ | $0.750^{+6.3\%}$ | 0.578 | $0.710^{+13.2\%}$ | $0.765^{+18.7\%}$ |
| Average | Accuracy↑ | 0.816 | N.A. | 0.589 | 0.759 | 0.743 | $0.815^{+7.2\%}$ | $0.846^{+10.3\%}$ | 0.742 | $0.806^{+6.4\%}$ | $0.847^{+10.5\%}$ |
| | Macro-F1↑ | 0.772 | N.A. | 0.469 | 0.624 | 0.652 | $0.724^{+7.2\%}$ | $0.768^{+11.6\%}$ | 0.604 | $0.710^{+10.6\%}$ | $0.771^{+16.7\%}$ |

imaging modalities, such as X-ray, OCT, ultrasound, CT, and electron microscopy, and supports various types of analysis, such as binary/multi-class classification, ordinal regression, and multi-label categorization, covering a total of 70 unique classification categories. The dataset comprises a total of 518,175 training samples, 70,467 validation samples, and 119,320 testing samples, covering a broad spectrum of diseases and classification types. For external validation, we employ the IDRiD Porwal et al. (2018), MESSIDOR Decencière et al. (2014), and APTOS Decencière et al. (2014) datasets. More dataset details are provided in Appendix. We report results using standard metrics such as accuracy and F1 score.

**Implementation Details.** In this work, we primarily evaluate our proposed method based on the 2B version of InternVL2 Chen et al. (2024) due to its effectiveness and efficiency, which demonstrates comparable or superior performance to other VLMs with larger parameter sizes in our experiments. InternVL2-2B consists of a ViT-Large vision encoder (InternViT-300M Chen et al. (2023b)) and a 1.8B-parameter language model (InternLM2-1.8B Cai et al. (2024)). During fine-tuning, we freeze the vision encoder and apply LoRA Hu et al. (2021) for efficient adaptation of the LLM component. Additionally, we introduce a novel vision-language connector specifically for the TSM model while keeping the TSM parameters fixed. For our VITask framework, we train stage 1 for 1 epoch, followed by stage 2 for an additional epoch.

**Compared Methods.** We compare our VITask-tuned VLM (VITask for short) against both a task-specific ViT classifier (TSM) and vanilla visual instruction-tuned VLMs on the MedMNIST dataset to analyze its task-specific performance, flexibility, and robustness. In particular, we test LLaVA1.5-

Table 2: Ablation study of the proposed components. RDA represents Response Distribution Alignment, CRT denotes Contrastive Response Tuning, and EP stands for Exemplar Prompting.

| Method | | Chest | | Derma | | OCT | | Retina | | Tissue | |
|---|---|---|---|---|---|---|---|---|---|---|---|
| | | Acc. | F1 | Acc. | F1 | Acc. | F1 | Acc. | F1 | Acc. | F1 |
| w/o EP | Vanilla | **0.523** | 0.024 | 0.770 | 0.499 | 0.726 | 0.704 | 0.590 | 0.370 | 0.569 | 0.419 |
| | +RDA | 0.517 | 0.078 | 0.799 | 0.585 | 0.844 | 0.837 | 0.615 | 0.401 | 0.632 | 0.523 |
| | +CRT | 0.513 | 0.088 | 0.786 | 0.593 | 0.817 | 0.810 | 0.593 | 0.413 | 0.622 | 0.505 |
| | +Both | 0.513 | **0.102** | 0.810 | 0.633 | 0.853 | 0.846 | 0.625 | 0.457 | 0.643 | 0.538 |
| w/ EP | Vanilla | 0.514 | 0.118 | 0.863 | 0.725 | 0.951 | 0.950 | 0.608 | 0.489 | 0.760 | 0.689 |
| | +RDA | 0.514 | 0.123 | 0.873 | 0.760 | 0.949 | 0.950 | 0.627 | 0.471 | 0.761 | 0.691 |
| | +CRT | 0.513 | 0.122 | 0.878 | 0.774 | 0.949 | 0.950 | 0.623 | 0.509 | 0.762 | 0.691 |
| | +Both | 0.517 | 0.129 | 0.877 | 0.772 | 0.952 | 0.952 | 0.632 | 0.522 | 0.761 | 0.690 |

13B Liu et al. (2023a), Qwen2-VL Bai et al. (2023), LLaVA-Med Li et al. (2024b) and InternVL2-2B Chen et al. (2024) with vanilla visual instruction tuning. For comprehensiveness, we also compare a recent medical VLM, MedDr He et al. (2024), which included MedMNIST as training set.

**Main Results.** Table 1 presents the medical image diagnosis performance across different models. *Comparison with TSM:* Most instruction-tuned VLMs, except VITask, show a significant performance gap compared to TSM, highlighting the challenges of fine-tuning VLMs for specialized tasks and domains. In contrast, VITask with Exemplar Prompting (EP) consistently delivers the best results, achieving the highest accuracy and F1 scores on 8 out of 12 datasets. This demonstrates that features derived from TSM are highly effective in providing VLMs with task-specific features, enabling VLMs to achieve TSM-level performance. Moreover, the superior performance of VITask relative to TSM suggests that it not only learns a good exemplar-response mapping but also leverages complementary information from both the pre-trained VLM and the TSM, offering enriched representations for maintaining basic conversation while excelling at specific tasks.

*Comparison with instruction-tuned VLMs:* Although MedDr performs well in some cases, this is likely due to its large size (26B parameters) and training on more medical datasets. Nonetheless, VITask with and without EP, despite having only 2B parameters, significantly outperforms MedDr on datasets like DermaMNIST, OCTMNIST, and OrganAMNIST. This further underscores the effectiveness of VITask in boosting task-specific performance. When comparing VITask to other VLMs tuned using vanilla visual instruction methods, its advantages become even more pronounced. VITask with and without EP outperforms LLaVA-13B, the second-best instruction-tuned VLM, by an average of $8.6\%$ and $14.7\%$ in F1 score, respectively. Furthermore, compared to InternVL-2B, which shares the same pre-trained VLM as VITask, our approach shows improvements in both accuracy and F1 score. This reinforces that VITask's enhancements are derived from its unique framework and strategies for task adaptation.

**Ablation Study.** In this section, we analyze the effectiveness of the three core components, exemplar prompting (EP), response distribution alignment (RDA), and contrastive response tuning (CRT), through ablation studies to understand their individual contributions to the overall performance. As shown in Table 2, when EP is disabled during inference, applying RDA improves the base model, InternVL-2B, by an average of $8.16\%$ in F1 score. Similarly, CRT alone improves the base model by $7.86\%$ in F1 on average. These results highlight that both RDA and CRT can independently boost task-specific performance. When RDA and CRT are combined, we observe additional improvements in both accuracy and F1 score, indicating that these two strategies complement each other to achieve optimal performance. When EP is used during inference, RDA does not yield notable gains. This is expected, as RDA is primarily designed to enhance performance in the absence of exemplars during inference. CRT, on the other hand, can still provide an improvement even with EP, but the margin of improvement is smaller. This is likely because the exemplar-prompted features have already adjusted the response distribution, reducing the necessity for further fine-tuning via CRT.

**Validation on External Datasets.** We further validate the external performance of instruction-tuned VLMs on the APTOS, IDRiD, and MESSIDOR datasets for diabetic retinopathy grading. These datasets use the same instruction formatting as RetinaMNIST but were not included during instruction

Table 3: Validation on external datasets.

| | ATPOS | | IDIRD | | Messidor | |
|---|---|---|---|---|---|---|
| | Acc. | F1 | Acc. | F1 | Acc. | F1 |
| TSM | 0.593 | 0.377 | 0.398 | 0.316 | 0.584 | 0.263 |
| Vanilla | 0.523 | 0.291 | 0.417 | 0.223 | 0.587 | 0.212 |
| VITask | 0.456 | 0.336 | 0.379 | 0.262 | 0.521 | 0.321 |
| VITask[plug] | **0.668** | **0.407** | **0.544** | **0.359** | **0.652** | **0.438** |

tuning. We evaluated the TSM, vanilla instruction-tuned VLM, and VITask w/ EP models, all of which were trained on RetinaMNIST. Additionally, we tested a variant of VITask, VITask[plug], which uses a newly trained TSM on the external datasets, replacing the original TSM for VITask without further fine-tuning. The results, as shown in Table 3, indicate that performance drops significantly for all models when tested on external datasets, highlighting the challenge of out-of-distribution generalization. As expected, the TSM, optimized for the specific task, achieves the best external performance. VITask is the second-best method, showing some generalization to external datasets. The vanilla VLM baseline achieved higher accuracy but lower F1 scores than VITask, likely due to the external datasets being biased with many normal cases, inflating accuracy. VITask[plug] outperformed other VLM-based methods, demonstrating VITask's flexibility in adapting to different tasks without the need for retraining the entire model.

**Robustness to Incomplete Instructions.** We also tested the robustness of instruction-tuned VLMs to incomplete instructions on the DermaMNIST dataset. We modified the dataset by removing references to possible disease names from the original instructions, eliminating necessary context information and making the instruction-following task more challenging. We then fine-tuned both the vanilla instruction-tuned VLM and VITask (with EP disabled for fairness) on this modified dataset. As illustrated in Figure 5, the vanilla visual instruction-tuned model's F1 score dropped dramatically from 0.531 to 0.423 when trained with incomplete instructions, showing that it heavily relies on detailed instructions for generating accurate responses. In contrast, VITask showed only a slight decrease in performance, demonstrating much better robustness against incomplete instructions. This resilience can be attributed to VITask's ability to implicitly align the VLM's response distribution with that of the TSM, providing a well-defined latent space that effectively characterizes desirable responses, even in the absence of detailed instructions.

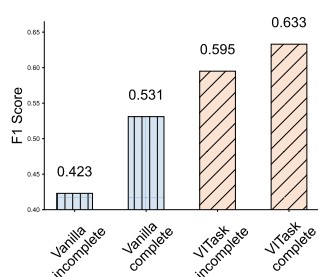

Figure 5: Robustness to incomplete instructions.

**Limitations and Discussions.** Our work has several limitations. Firstly, we primarily focus on image classification tasks, where training a single TSM for all tasks is straightforward. However, for other instruction-following tasks, such as image captioning and VQA, training such a TSM may not be as feasible or effective. Extending the VITask framework to these types of tasks remains a challenge and could be an avenue for future research. Secondly, our experiments are limited to medical datasets. While the results demonstrate the effectiveness of VITask in the medical domain, testing across a broader range of domains would be necessary to fully validate its generalizability. Exploring VITask's applicability to datasets beyond the medical field is an important next step. Lastly, we focus on task-specific training during the fine-tuning stage. However, we believe that our method has the potential to enhance both the pre-training and fine-tuning phases of VLMs to achieve task-specific model-level performance. Exploring VITask's application to pre-training could lead to further improvements in adaptability and performance across diverse tasks.

## 6 CONCLUSION

In this paper, we proposed VITask, a novel framework that bridges task-specific models (TSM) and visual language models (VLM) to enhance task-specific adaptability and performance. Through exemplar prompting (EP), response distribution alignment (RDA), and contrastive response tuning (CRT), VITask leverages specialized task features from TSMs and aligns them with the instruction-following capabilities of VLMs. Our experiments demonstrate that VITask outperforms both conventional instruction-tuned VLMs and TSMs across a variety of datasets, showcasing its ability to integrate complementary features from both models effectively. VITask not only improves task-specific performance but also introduces practical advantages, such as flexibility in incorporating any TSM architecture in a plug-and-play manner, and robustness to incomplete instructions. By decoupling image representation learning from instruction tuning, VITask offers an efficient and adaptable solution for new and unseen tasks without the need for extensive retraining.

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
