# A APPENDIX

## A.1 VITASK PERFORMANCE ON ORIGINAL TASK.

To verify that VLM maintains its original capabilities while adapting to new tasks via our VITask fine-tuning, we evaluated different fine-tuning approaches on eight benchmark datasets originally used to assess the pre-trained InternVL2-2B model: MMBench_en Liu et al. (2025), CCBench Liu et al. (2025), POPE Li et al. (2023c), MMMU Yue et al. (2024b), MMVP Tong et al. (2024), MMVet Yu et al. (2023b), GQA Hudson & Manning (2019), and AI2D Kembhavi et al. (2016). As shown in Table 4, our method preserves the original capabilities of the base model across all benchmarks, with differences typically within 3% from the original model (e.g., MMBench_en: 72.2 vs. 73.2, CCBench: 72.5 vs. 74.7). In contrast, alternative fine-tuning approaches that make either the vision encoder or both the vision encoder and connector learnable during fine-tuning suffer from severe performance degradation, with accuracy dropping by up to 90% on some benchmarks.

Table 4: Performance comparison of InternVL2-2B before and after fine-tuning.

| Method | MMBench↑ | CCBench↑ | POPE↑ | MMMU↑ | MMVP↑ | MMVet↑ | GQA↑ | AI2D↑ |
|---|---|---|---|---|---|---|---|---|
| Original | 73.20 | 74.7 | 87.3 | 0.342 | 0.353 | 44.6 | 61.03 | 0.741 |
| Tunning LLM + Vision | 27.60 | 27.7 | 0.1 | 0.274 | 0.112 | 15.5 | 30.18 | 0.021 |
| Tunning LLM + Vision + Conn | 17.15 | 8.0 | 0.7 | 0.291 | 0.073 | 13.0 | 27.80 | 0.005 |
| Ours | 72.20 | 72.5 | 87.7 | 0.341 | 0.340 | 36.2 | 59.60 | 0.700 |

## A.2 FURTHER ABLATIONS ON THE EFFECTIVENESS OF EXEMPLAR PROMPTING.

To thoroughly investigate the effectiveness of Exemplar Prompting (EP), we conducted several ablation studies as shown in Table 5. The results show that EP's performance gain stems from the specialized features of the external TSM rather than additional learnable parameters, as evidenced by consistently outperforming baselines like LLM+Conn (tune LLM and connector parameters) and LLM+Conn+Prefix (tune LLM, connector and additional learnable prefix parameters) across medical tasks. Further experiments reveal that while fine-tuning the vision encoder (LLM+Vision) can improve downstream task performance, it causes catastrophic forgetting of pre-trained knowledge, making it impractical for maintaining generalizability.

Table 5: Classification performance (Acc./F1) of InternVL2-2B with different fine-tuning approaches (**Best**, Second Best).

| Model | Path | Chest | Derma | OCT | Pneumonia | Retina | Breast | Blood | Tissue | OrganA | OrganC | OrganS |
|---|---|---|---|---|---|---|---|---|---|---|---|---|
| Baseline | 0.926/0.896 | **0.523**/0.024 | 0.770/0.499 | 0.726/0.704 | 0.886/0.873 | 0.590/0.370 | 0.744/0.524 | 0.931/0.818 | 0.569/0.419 | 0.828/0.801 | 0.778/0.742 | 0.635/0.578 |
| LLM+Conn | 0.940/0.912 | 0.511/0.078 | 0.773/0.497 | 0.808/0.798 | 0.905/0.893 | 0.590/0.359 | 0.782/0.658 | 0.975/0.856 | 0.617/0.502 | 0.898/0.886 | 0.862/0.838 | 0.728/0.679 |
| LLM+Conn+Prefix | 0.941/0.913 | 0.513/0.078 | 0.775/0.503 | 0.813/0.803 | 0.920/0.910 | 0.608/0.384 | 0.808/0.724 | 0.973/0.855 | 0.616/0.497 | 0.905/0.891 | 0.863/0.837 | 0.724/0.670 |
| LLM+Vision | **0.972/0.964** | 0.510/**0.134** | 0.835/0.658 | 0.891/0.891 | 0.910/0.899 | 0.598/0.425 | 0.865/0.828 | 0.986/0.867 | 0.738/0.659 | **0.962/0.960** | 0.932/0.919 | 0.824/0.777 |
| LLM+Vision+Conn | 0.967/0.957 | 0.511/0.127 | 0.822/0.641 | 0.898/0.897 | 0.923/0.914 | 0.605/0.418 | 0.859/0.815 | 0.990/0.869 | 0.736/0.655 | **0.963/0.959** | **0.935/0.923** | **0.825/0.781** |
| EP | 0.948/0.931 | 0.514/0.118 | **0.863/0.725** | 0.951/0.950 | 0.941/0.935 | 0.608/0.489 | 0.878/0.836 | 0.991/0.870 | 0.760/0.689 | 0.951/0.942 | 0.894/0.885 | 0.788/0.747 |
| EP+Vision | 0.645/0.660 | 0.241/0.053 | 0.748/0.356 | 0.937/0.937 | 0.716/0.605 | 0.345/0.270 | 0.795/0.763 | 0.985/0.863 | 0.672/0.512 | 0.918/0.913 | 0.828/0.805 | 0.739/0.672 |

## A.3 VITASK PERFORMANCE ON NATURAL IMAGE DOMAIN.

To verify VITask's effectiveness beyond medical images, we evaluated our method on three natural image classification datasets: Stanford Cars Gebru et al. (2017), Flowers 102 Nilsback & Zisserman (2008), and Caltech 101 Fei-Fei et al. (2006). As shown in Table 6, VITask significantly improves the accuracy of the vanilla-tuned InternVL2-2B model (e.g., from 77.4% to 85.4% on Stanford Cars and from 89.9% to 99.0% on Flowers 102), achieving results comparable to the specialized TSM (86.2%, 99.2%, and 97.6% respectively). These results demonstrate that our method is effective and broadly applicable across both general and medical domains.

## A.4 DATASET AND INSTRUCTION PROMPT.

We utilize the **MedMNIST** dataset collection Yang et al. (2023) as our primary training and testing dataset for VLM, which comprises 12 distinct 2D datasets. Detailed descriptions of each dataset are provided below:

Table 6: Classification performance (Acc.) of the fine-tuned VLM on natural image datasets.

| Method | Stanford Cars | Flowers 102 | Caltech 101 |
|---|---|---|---|
| TSM | 0.862 | 0.992 | 0.976 |
| InternVL2-2B | 0.774 | 0.899 | 0.960 |
| InternVL2-2B + Ours | $0.854^{+8.0\%}$ | $0.990^{+9.1\%}$ | $0.980^{+2.0\%}$ |

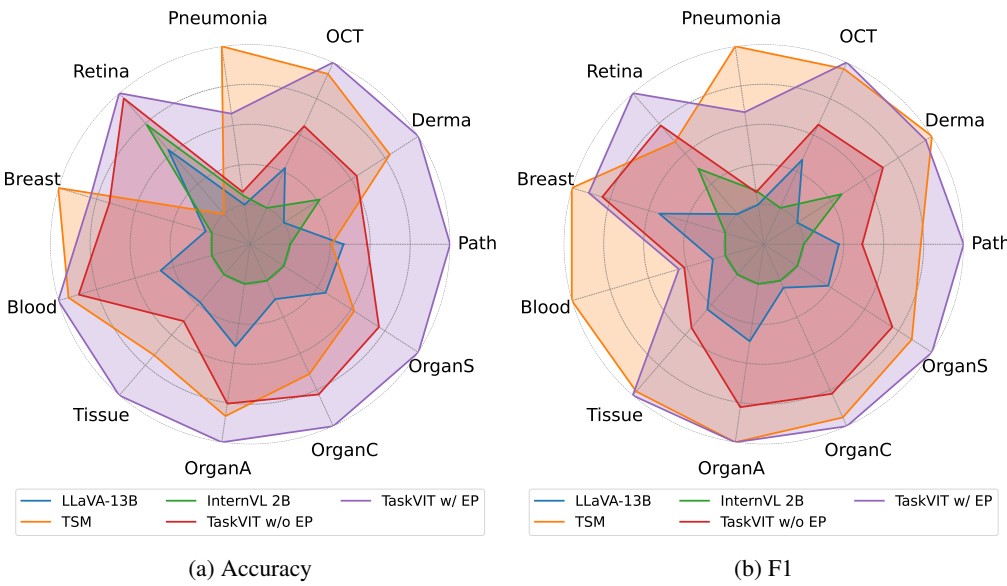

(a) Accuracy         (b) F1

Figure 6: Performance of VITask in adapting to different tasks.

- **PathMNIST** Kather et al. (2019): Derived from the NCT-CRC-HE-100K dataset based on colorectal cancer histology slides, this dataset includes $100,000$ training image patches and $7,180$ test patches from a different clinical center, classified into 9 tissue types for multi-class classification.

- **ChestMNIST** Kermany et al. (2018): Based on the NIH-ChestXray14 dataset, it comprises $112,120$ frontal-view chest X-ray images of $30,805$ unique patients, labeled with 14 disease categories for multi-label classification.

- **DermaMNIST** Tschandl et al. (2018); Codella et al. (2019): Sourced from the HAM10000 dataset, a large collection of multi-source dermatoscopic images, it contains $10,015$ images categorized into 7 different skin conditions for multi-class classification.

- **OCTMNIST** Kermany et al. (2018): Derived from a prior dataset on retinal optical coherence tomography (OCT) images, it comprises $109,309$ samples categorized into 4 diagnostic classes for multi-class retinal disease classification.

- **PneumoniaMNIST** Kermany et al. (2018): Based on a collection of pediatric chest X-ray images, it includes $5,856$ samples for binary classification of pneumonia against normal cases.

- **RetinaMNIST** Liu et al. (2022): Developed from the DeepDRiD challenge dataset, this collection includes $1,600$ retina fundus images labeled for 5-level diabetic retinopathy severity and formulated as an ordinal regression task.

- **BreastMNIST** Al-Dhabyani et al. (2020): Sourced from a dataset of $780$ breast ultrasound images, it is categorized into 3 classes—normal, benign, and malignant—and simplified into binary classification for the current study.

- **BloodMNIST** Acevedo et al. (2020): This dataset features $17,092$ images of individual normal blood cells, categorized into 8 classes based on cell type, for multi-class classification.

- **TissueMNIST** Ljosa et al. (2012): Developed from the BBBC051 dataset, it contains $236,386$ human kidney cortex cell images segmented into 8 tissue types for classification.
- **Organ{A,C,S}MNIST** Bilic et al. (2023); Xu et al. (2019): Sourced from the Liver Tumor Segmentation Benchmark (LiTS) dataset, it contains 2D images obtained from 3D CT scans of 11 body organs. The dataset is split into three separate views (axial, coronal, and sagittal), each forming a multi-class organ classification task.

The diverse array of datasets provides a solid foundation for evaluating our method across multiple biomedical imaging domains, supporting both binary and multi-class classification tasks.

We construct the instruction-response pairs for medical image classification following the approach outlined in previous work He et al. (2024). Specifically, to prepare the image classification data for ViTask training and testing, each dataset is converted into an instruction-tuning format by rephrasing the classification task as a question about the disease observed in the image, along with a set of possible disease options. The response corresponds to the correct disease name. The data construction template is shown below.

**User:**
```
Analyze the given {Modality} image.
The possible diagnoses are:{Label Set}.
```
**VITask:**
```
{Label}.
```

### A.5    IMPLEMENTATION DETAILS.

**Training of Task-Specific Model for Exemplar Prompting.** For the task-specific model used in Exemplar Prompting, we employ a ViT Dosovitskiy et al. (2020) base model pre-trained on ImageNet-21K Deng et al. (2009), and fine-tune it on the MedMNIST dataset for training, testing, and validation. We combine all 2D datasets in MedMNIST and jointly train the ViT model across all 70 classification tasks (i.e., using a shared classification head with 70 classes). During training, the loss is computed only over the class subset corresponding to the current sample's dataset. We train the ViT model for 30 epochs and select the best model based on validation set performance.