# OpenReview forum: "From Generalist to Specialist: Adapting Vision Language Models via Task-Specific Visual Instruction Tuning"
_ICLR.cc/2025/Conference — Submitted to ICLR 2025_

### Official Review · Reviewer_Z7hL · 2024-10-29

**Soundness:** 3
**Presentation:** 4
**Contribution:** 3
**Rating:** 6
**Confidence:** 4

**Summary:**

This paper first investigate why fine-tuned (VLMs) often lag behind TSMs in performance (use image classification as a case study) and then propose a new framework that can enhance the performance of VLMs on task-specific applications. They conduct it through integrating task-specific models (TSM) into VLMs in fine-tuning with 3 strategies, exemplar prompting (EP), response distribution alignment (RDA), and contrastive response tuning (CRT). EP allow TSM features to guide VLMs while RDA enables VLMs to adapt without TSMs during inference by learning from exemplar-prompted models. CRT further optimizes the ranking of correct image-response pairs by maximizing likelihood of correct image-response pairs while minimizing that of  incorrect pairs. They conduct two-stages training. The first stage is fine-tuning VLMs using vanilla visual instruction tuning in conjunction with EP and RDA. The second stage is fine-tuning VLMs using all strategies. The experiments focus on classification tasks on medical domain.

**Strengths:**

- The idea and strategies used to combine TSMs to VLMs to boost performance of VLMs (without using TSMs to VLMs on inference stage based on RDA mechanism) on task-specific applications are interesting. It helps to remove the limitation of VLMs when fine-tuning on task-specific applications.
- The performance of experiments on medical image classification are good.
- The paper is well-written and easy to follow.

**Weaknesses:**

- The current experiment results are good, but the number of experiments and baseline models are insufficient, only 4 baseline models (including task-specific model). Can you compare your method with more other baselines, including some works you have mentioned in related works section which attempt to integrate VLMs with task-specific models and more VLMs such as InstructBLIP, BLIP-2, and Qwen-VL-Chat?
- Because you did classification experiments on medical domain and there are currently many VLMs pre-trained on medical datasets, why you chose to test on the particular VLMs used in the study. In my opinion, it is important to test this approach on more VLMs which were pre-trained on medical datasets such as LLaVA-Med, Biomed GPT, etc. to make sure applying this method on these VLMs give significant improvement results.
- I am concerned about the time used to fine-tune a VLM using your method, because you need to fine-tune a task-specific model on the dataset first before combining it with VLMs and then fine-tune the VLMs. Can you give me more detail about the resource you use to fine-tune VLMs and the total time for the training process to finish using your method and vanilla methods (comparisons between your method and baseline approaches) ?

**Questions:**

As you have mentioned in the paper, testing this approach only on medical domain is not enough to make sure your method is good. I hope that you can give more experiment results on other non-medical domains such as natural images.

---

> ### Author Response · Authors · 2024-11-26
> **Response to Reviewer Z7hL**
>
> **1. Can you compare your method with more baseline models? Why you chose to fine-tune particular VLMs?**
>
> We expanded our comparisons to include additional baseline models such as LLaVA-7B v1.5, InternVL2-8B, and Qwen2-VL-7B-Instruct. As shown in **Table 2**, fine-tuned Qwen2-VL-7B and other models with standard tuning generally achieve comparable or worse performance than fine-tuned InternVL2-2B in most cases. Considering InternVL2-2B's smaller size and robust performance, we selected it as our base model for experimentation.
>
> **2. Verification of the proposed method on medical VLMs.**
>
> We applied our method to LLaVa-Med v1.5 to assess its effectiveness on medical VLMs. As shown in **Table 2**, our method improves LLaVa-Med's accuracy and F1 score by 10.3% and 11.6%, respectively, demonstrating its capability to enhance both general and medical VLMs.
>
> Additionally, we verified that our method preserves the base VLM's original capabilities while enabling it to handle new tasks. As shown in **Table 1**, the InternVL2-2B achieves comparable performance on its original benchmarks before and after fine-tuning, confirming that our method generalizes beyond task-specific specialization.
>
> **3. Concerns about fine-tuning time and resources used.**
>
> In our experiments, training a TSM for classification with ~500K samples takes 2.5 hours on 4 Nvidia A100 GPUs. Standard visual instruction tuning of InternVL2-2B takes 2 hours, while our VITask method takes 4 hours to fine-tune the same base VLM.
>
> Given the substantial task-specific performance improvements, we consider the computational cost of training a TSM and applying VITask acceptable. Moreover, training the TSM from scratch on the same dataset is done purely for fair comparison in this study but is not always necessary in practice. Our method is flexible and can leverage pre-existing models as TSMs, which will significantly reduce computational cost.
>
> To demonstrate this flexibility, we utilized SAM-Med2D [12] as the TSM for medical phrase grounding. Results in **Table 3** show that our method improves Recall@0.5 by 4.63% on average, confirming its flexibility and effectiveness across tasks, including classification and phrase grounding.
>
> **4. Validation on non-medical domains.**
>
> Thank you for the suggestion. We conducted additional experiments on three natural image classification datasets under a challenging open-world setting, where no candidate answers are provided in the instruction. Results in **Table 4** demonstrate that our method significantly enhances the fine-tuned VLM's performance, achieving results comparable to the TSM, which represents the upper bound for classification accuracy. These findings validate that our method is effective and broadly applicable across both general and medical domains.

---

> ### Comment · Reviewer_Z7hL · 2024-11-26
>
> Thank you for your rebuttal. Most of my concerns are addressed and the overall experiment results on medical domain are impressive.
>
> However, I concern about the results on natural images. The performance when using TSM and your method are quite equal, which do not show significant improvement of your method compared to TSM. I think because TSM have so good performance on these dataset, which make the result do not show any significant improvement.
>
> In addition, your method now only focus on classification task and I think there are many tasks in medical domain which are more important than classification such as VQA, Medical Report, so I keep my original score.

---

> > ### Author Response · Authors · 2024-11-30
> > **Follow-Up on Reviewer Feedback**
> >
> > Dear Reviewer **Z7hL**,
> >
> > Thank you for reviewing our rebuttal and acknowledging the improvements made. We greatly appreciate your detailed feedback and thoughtful comments. Below, we address your remaining concerns:
> >
> > **1. Performance on natural images.**
> >
> > We acknowledge that our method does not outperform the TSM, which is **expected** since the TSM backbone benefits from extensive pre-training on natural images. However, our method significantly improves the performance of the **fine-tuned VLM** using *standard instruction tuning on the same training data*, narrowing the gap to **match TSM-level performance**. This demonstrates that our approach effectively adapts VLMs to achieve comparable performance to specialized models.
> >
> > **2. Applications of our method.**
> >
> > In **Table 3**, we demonstrated the generalizability of our method beyond classification through its application to *medical phrase grounding*. For other medical tasks, such as VQA and report generation, we will explore adaptations of our method as part of future work.
> >
> > We welcome any further suggestions or concerns and are committed to addressing them thoroughly before the review process concludes.
> >
> > Thank you again for your time and constructive feedback.
> >
> > Best regards,
> > **The Authors**

---

### Official Review · Reviewer_YD9v · 2024-11-03

**Soundness:** 3
**Presentation:** 3
**Contribution:** 3
**Rating:** 8
**Confidence:** 4

**Summary:**

Although VLMs have excelled in understanding generic images, they fall apart for out-of-distribution task-specific applications. This paper claims that this could be due to the lack of comprehensive image representation and indirect tuning objective VLMs used to optimize. This paper proposes a framework that leverages representation from task-specific models as complementary information and optimizes the VLMs. Additionally, it explores how the inference can be done without task-specific models using two proposed methods RDA and CRT.
Considering the high costs of fine-tuning and inference with LLMs, along with the minimal performance gains, why should a user prefer using VITask over enhancing TSMs?

The paper is well-written but could be improved for better readability.

**Strengths:**

The motivation of the paper is clear and the authors provided adequate evidence to show why the proposed method can be useful in task-specific applications.

The task-specific instruction tuning section is quite interesting and the papers validate how each proposed component improves the performance of VLMs.

The paper has a detailed section providing intuition and empirical evidence to support the claims.

**Weaknesses:**

As mentioned in the limitations section, exploration of the method is highly limited to classification tasks. Additional tasks would be more interesting using an LLM, which has rich information embeddings.

Given the high computation budget and a marginal improvement in performance compared to TSMs, I would try to improve the TSMs which are cost-effective for training and inference. ( I consider the results as marginal improvement, the considered dataset is comparably simpler than the actual applications.) For example, optimize the model architecture, focused hyperparameter selection, Select and Fuse futures, and an ensemble of smaller models.

The paper is comprised of a lot of detailed text, which also seems to be a weakness of the paper. Authors could provide more visualization to interpret the results and behaviour of each module. Overall, the interpretation of results could be improved for better readability. You may provide more visualization to explain the benefits of the proposed modules or interpretation analysis how the proposed module benefits  compared to TSMs or vanilla VLMs.

The paper presents a few concerns :

1. The figure related to the RDA analysis referenced in Lines 264-266 is missing. As it stands, the current figure appears to be a bar chart of the EP results. Could please provide the figure?

2. The authors should consider using different formatting styles for equations (3-6). When the terms in the equations are presented in the same style as the surrounding text, it can be confusing (Line 280). Differentiating the formatting will enhance readability.

3. There are spelling errors in Figure 1 (d).

4. The word "often" is repeated in Line 041.

**Questions:**

1. Authors should explain why using Vision-Language Models (VLMs) with a high computational budget is more beneficial than improving Task-Specific Models (TSMs). I would recommend an empirical analysis of the cost and performance benefits of VITask versus TSMs to.

2. TSMs provide better task-specific representations than general vision encoders. Could you clarify why the results decline significantly when embeddings are completely replaced by task-specific representations? Did you utilize the CLS token from Vision Transformers (ViT) or the entire patch embeddings?

3. Why do we need to optimize Stage 2 with the same loss functions used in Stage 1 without scaling them? After optimizing the Stage 1 model to minimize the Stage 1 loss, using it in Stage 2 could potentially harm the fine-tuning process. Have you considered applying a scale to the Stage 1 loss functions?

4. In line 423, it is mentioned that a novel vision-language connector is introduced in the paper, but no details about this connector are provided in the text. Could you please include more information about it?

---

> ### Author Response · Authors · 2024-11-26
> **Response to Reviewer YD9v**
>
> **1. Evaluation on tasks beyond classification.**
>
> We appreciate the suggestion. To assess whether our method generalizes to tasks other than classification, we tested it on medical phrase grounding using SAM-Med2D [12] as the TSM on the Med-GRIT-270 dataset [11]. This dataset includes 270k question-answer pairs across eight medical imaging modalities and evaluates bounding box predictions for given textual findings.
>
> As shown in **Table 3**, our VITask method improves the Recall@0.5 by 4.63% on average compared to standard instruction tuning. These results demonstrate the effectiveness of VITask in enhancing the grounding ability of fine-tuned VLMs, suggesting its potential for a wide range of vision-language tasks.
>
> **2. Why use VLMs with a high computational budget instead of improving TSMs?**
>
> We acknowledge that directly improving a TSM is an effective approach for optimizing specific downstream tasks. However, fine-tuning a VLM aligns better with the broader goal of creating an AI agent capable of handling multiple tasks, as VLMs offer superior generalizability and user-friendliness.
>
> Our work emphasizes teaching a pre-trained VLM to handle new tasks while preserving its original capabilities, rather than optimizing for state-of-the-art performance on individual tasks. Using our method, the fine-tuned VLM acquires new capabilities without compromising existing ones.
>
> To validate this, we used InternVL2-2B as the base model and evaluated its performance before and after fine-tuning with our method on benchmark datasets originally designed for evaluating InternVL2. As shown in **Table 1**, our method introduces only minor degradation, confirming that the VLM's original capabilities are preserved.
>
> **3. Improving interpretation of results for better readability.**
>
> We appreciate your feedback. The comparisons between our methods and baseline models have been revised for clarity and better readability in the updated manuscript.
>
> **4. Missing figures, equation formatting (3–6), and typos.**
> Thank you for pointing these out. We have corrected the formatting issues, added the missing figures, and fixed typos throughout the paper.
>
> **5. Empirical analysis of VITask vs. TSMs in cost and performance.**
> Training the TSM for classification on 500K samples takes 2.5 hours using 4 Nvidia A100 GPUs, while the standard visual instruction tuning of InternVL2-2B takes 2 hours. Our VITask method requires 4 hours to fine-tune the same base VLM, which improving the F1 score of vanilla-tuned InternVL-2B from 60.4% to 77.1% and outperforming the TSM by 3.1% in average accuracy.
>
> We consider this computational cost acceptable given the significant performance improvement. Moreover, VITask is flexible and can utilize pre-trained TSMs, avoiding the need to train one from scratch. For instance, in medical phrase grounding, we use SAM-Med2D [12] as the TSM. As shown in **Table 3**, VITask is effective even for tasks beyond classification.
>
> The optional RDA strategy adds 2 hours of training but removes the reliance on TSMs during inference. RDA resolves scalability challenges when dealing with multiple tasks and numerous TSMs, which would otherwise require substantial storage and computation. With RDA, the fine-tuned VLM operates independently of TSMs while maintaining strong performance, offering a trade-off between test-time efficiency and effectiveness.
>
> **6. Why do results decline significantly when embeddings are completely replaced by task-specific representations? Did you use the CLS token or entire patch embeddings from ViT?**
>
> Task-specific features are not well-aligned with the LLM, which is pre-trained to process original vision embeddings. Replacing these embeddings entirely disrupts the generative capabilities of the VLM, leading to a significant drop in performance.
>
> In our method, we use the CLS token from the TSM as the feature for exemplar prompting. While we also tested using the entire patch embeddings, the results showed no substantial improvement. Therefore, we chose the CLS token for better efficiency.
>
> **7. Why use Stage 1 loss functions without scaling in Stage 2 optimization?**
>
> Thank you for pointing this out. In Stage 2 training, we downscaled the CRT weight by setting the hyperparameter $\beta = 0.1$, ensuring all loss terms are in a comparable range. This adjustment prevents any loss term from dominating, maintaining balance in the fine-tuning process.
>
> **8. What is the "novel" vision-language connector in Line 423?**
>
> We apologize for the confusion. The word "novel" was a typo and should be removed. The TSM connector is a simple MLP, similar to the one used in the original VLM.

---

> > ### Comment · Reviewer_YD9v · 2024-11-29
> >
> > Thank you to the reviewers for preparing a comprehensive rebuttal in such a short time. The rebuttal is well-structured and clear. I can see how your method generalizes and enhances the results when integrated with existing VLMs. I suggest including these results in the paper, as they highlight the significance of your methods. I'm curious about the reasons behind the steep drop in performance observed in Table 1 following fine-tuning. Did you use PEFT or direct weight adjustment? I feel more positive about this paper after reading the rebuttal.

---

> ### Author Response · Authors · 2024-11-30
> **Follow-Up on Reviewer Feedback**
>
> Dear Reviewer **YD9v**,
>
> Thank you for revisiting our work and for your positive assessment. We deeply appreciate your acknowledgment of the improvements made and the constructive feedback provided. Below, we address your follow-on questions:
>
> **1. Inclusion of results in the paper.**
>
> We completely agree that the results shared in the rebuttal are essential to demonstrating the significance of our method. We will incorporate these results and their accompanying discussions into the final version of the paper. We believe this addition will significantly enhance the clarity and impact of our work.
>
> **2. Reasons behind the steep drop in performance in Table 1.**
>
> For the LLM component of the VLM, we always use **LoRA** for fine-tuning. For the vision encoder and connector, we directly fine-tune their weights **without employing any PEFT techniques**. This direct weight adjustment explains the observed steep drop in performance following fine-tuning.
>
> We sincerely thank you for your thoughtful suggestions and insights. Please do not hesitate to reach out if you have further questions or additional feedback.
>
> Best regards,
> **The Authors**

---

### Official Review · Reviewer_rqyz · 2024-11-04

**Soundness:** 3
**Presentation:** 3
**Contribution:** 2
**Rating:** 6
**Confidence:** 4

**Summary:**

This paper proposes a combination of strategies, exemplar prompting (EP), response distribution alignment (RDA), and contrastive response tuning (CRT), to adapt general-purpose generative VLMs for specialized medical classification tasks. Through experiments, the authors show that their method VITask , leveraging these three algorithmic components, can outperform task-specific models (TSMs), which are essentially fine-tuned ViTs, on a variety of medical classification tasks. In ablation experiments, the authors demonstrate that EP contributes the most performance boost while RDA and CRT are also complementary.

**Strengths:**

The paper states the problem it aims to solve clearly. The proposed method is validated on a wide range of medical datasets.

**Weaknesses:**

The biggest issue of the paper is the lack of depth. While it ablates the impact of each of the algorithmic components, they authors spent little effort trying to understand why each of them work and to compare them against existing methods.

1. It’s not clear what makes EP successful.
    - I strongly suspect the performance gain is mostly due to the fine-tuning of the connector module. The critical experiment of simply having both the connector and the LLM (LoRA params) trainable is missing.
    - Additionally, an experiment comparing EP with prefixing tuning [1] will tell whether it’s necessary to condition the prefix (additional tokens to the LLM’s embedding space) on the image at all to get good performance. Essentially, I need to see experiments showing me EP > fine-tuning the original VLM’s connector + prefix tuning to be convinced it’s novel.
    - I also don’t buy the claim that fine-tuning the Vision model in VLM will distort vision language alignment at all. If fine-tuning the Vision model is harmful, wouldn’t the trained LoRA weights be more harmful as well? A controlled experiment where the vision encode is also trained is needed. I am confident this will make EP perform even better.
    - Finally, other works with the same core methodology should be discussed. For example, Graph Neural Prompting [2] builds a knowledge graph based on the prompt and multiple choice candidates and generates a conditional prefix to prompt the LLM. I think the idea is extremely similar to EP.
2. Regarding RDA: this is essentially a fancy way of saying knowledge distillation but no relevant papers are cited. Regarding implementation, the author mentions gradient detachment. If I understood it correctly, this just means the TSM, or the “teacher”, is not trained while the goal is to train the student. Shouldn’t this be the default setting anyway?
3. Contrastive Response Tuning: as part of the core methodology, the paper should compare its effectiveness against existing methods, such as contrastive decoding [3][4].

Issues mentioned above should be addressed. Otherwise this work should aim for a more application-oriented venue.

The notations issues.

- In equations (1), (2), (3), (5), (6), why is there a min() operator on the left hand side? The author seems to mix it up with the argmin notation. I think the author should remove the min() and avoid argmin() like notation since not all parameters are trained.

Minor grammar issues
- For example, Takeaway #1: TSM features can prompts (prompt) VLMs to generate desired responses.

References:

[1] Li, Xiang Lisa, and Percy Liang. "Prefix-Tuning: Optimizing Continuous Prompts for Generation." Proceedings of the 59th Annual Meeting of the Association for Computational Linguistics and the 11th International Joint Conference on Natural Language Processing (Volume 1: Long Papers). 2021.
[2] Tian, Yijun, et al. "Graph neural prompting with large language models." Proceedings of the AAAI Conference on Artificial Intelligence. Vol. 38. No. 17. 2024.
[3] Leng, Sicong, et al. "Mitigating object hallucinations in large vision-language models through visual contrastive decoding." Proceedings of the IEEE/CVF Conference on Computer Vision and Pattern Recognition. 2024.
[4] Favero, Alessandro, et al. "Multi-modal hallucination control by visual information grounding." Proceedings of the IEEE/CVF Conference on Computer Vision and Pattern Recognition. 2024.

**Questions:**

1. For TSM, does it use the same encoder as the one used by the target VLM? How about the connector?
2. What parameters are tuned in RDA exactly? I also don’t get how is it really different from tuning from the ground truth labels, which are used anyway. In other words, isn’t p_theta(response | exemplar) just an approximation of the ground truth label? In fact, this pseudo-label is worse since it can be wrong? Also for classification tasks, if the model only decodes a single label, does it really matter to learn the whole distribution instead of an one-hot label? If we look at table 2, RDA indeed improves upon vanilla, but what does vanilla mean? Is it just the base VLM that’s not fine-tuned at all for those medical tasks? I need to see an experiment where you just fine-tune with the ground truth labels in the training set to really understand the role of RDA. Now I tend to think you can totally get rid of it.
3. For both RDA and CRT, what did you do to deal with the situation when the label consists of more than 1 token due to tokenization?
4. I want to know the implementation detail of how you extract the answer from the generated output from the VLM. Apparently since it’s free-form generation the format might be slightly off. Are there any instances where the model outputs a semantically equivalent label that’s not an exact match? What’s the percentage of such instances before and after your fine-tuning?
5. I wonder if the authors can explain more on the motivation of adapting the VLM for classification tasks. I think this task is only meaningful if either 1) the original generative capabilities can be preserved, or 2) the fine-tuned VLM is actually better than SoTA vision models, otherwise we might as well use a predictive classifier that’s smaller and easier to fine-tune. Thus, I think the authors should either 1) present evaluations on non-medical tasks to show there’s very little degradation, or 2) compare against the SoTA vision model, which is probably not a ViT pretrained in 2020.

---

> ### Author Response · Authors · 2024-11-26
> **Response to Reviewer rqyz (1/4)**
>
> Thank you for your detailed comments and valuable suggestions, which have greatly improved the quality of our paper. Below, we address your concerns and clarify potential misunderstandings:
>
> **1. Justification of what makes EP successful.**
>
> **1.1 Does the performance gain of EP come from fine-tuning additional learnable model parameters?**
>
> The performance improvement of EP stems from the specialized features from the external TSM, rather than the fine-tuning of additional learnable modules such as the connector between the TSM and LLM.
>
> To validate this, we conducted additional experiments comparing several baselines:
> - Standard LoRA tuning (Baseline)
> - Fine-tuning the original VLM connector and LoRA parameters of LLM (LLM+Conn)
> - LLM+Conn with prefix tuning (LLM+Conn+Prefix)
>
> **Table 5** shows that EP >> LLM+Conn+Prefix > LLM+Conn > standard tuning. Specifically, fine-tuning the VLM connector improves performance to some extent, as expected. Adding prefix tuning further enhances LLM+Conn, albeit marginally. However, both LLM+Conn and LLM+Conn+Prefix consistently underperform compared to EP, often by a significant margin. These results demonstrate that EP's performance gain is not attributed to fine-tuning additional learnable parameters.
>
>
> **1.2 Why does fine-tuning the vision model in VLM distort vision-language alignment? If so, wouldn’t trained LoRA weights also be harmful?**
>
> Fine-tuning the vision model in a VLM distorts vision-language alignment in the sense that it alters the entire latent space of image embeddings, causing the VLM to forget pre-trained knowledge and tasks. However, this does not imply that fine-tuning the vision model is harmful for learning new downstream tasks. In comparison, instruction-tuned LLMs are more robust to forgetting because their tuning is conditioned on new instructions, which has less impact on their pre-trained instruction-following abilities.
>
> To validate this, we compared EP with the following baselines:
> - Fine-tuning the vision encoder and LoRA parameters of LLM (LLM+Vision)
> - Fine-tuning both the vision encoder and connector with LoRA parameters (LLM+Vision+Conn)
> - EP with a learnable vision encoder (EP+Vision)
>
> As shown in **Table 5**, making the vision encoder learnable significantly improves downstream task performance. However, further tuning the connector does not yield additional benefits, and EP still outperforms or performs on par with VLM+Vision and VLM+Vision+Conn. Notably, EP with a learnable vision encoder does not lead to better performance. These findings suggests that learning from the TSM features is more effective than tuning the original vision encoder.
>
> Furthermore, as will be discussed later, fine-tuning the vision encoder causes the VLM to completely forget pre-trained knowledge, which is impractical for generalizability. These results further highlight the advantages of EP.
>
> **Table 5. Classification performance (Acc/F1) of InternVL2-2B with different fine-tuning approaches.**
> | Model                        | Path          | Chest         | Derma        | OCT          | Pneumonia    | Retina       | Breast       | Blood        | Tissue       | OrganA       | OrganC       | OrganS       |
> |------------------------------|---------------|---------------|--------------|--------------|--------------|--------------|--------------|--------------|--------------|--------------|--------------|--------------|
> | Baseline | 0.926/0.896 | **0.523**/0.024 | 0.770/0.499 | 0.726/0.704 | 0.886/0.873 | 0.590/0.370 | 0.744/0.524 | 0.931/0.818 | 0.569/0.419 | 0.828/0.801 | 0.778/0.742 | 0.635/0.578 |
> | LLM+Conn | 0.940/0.912 | 0.511/0.078 | 0.773/0.497 | 0.808/0.798 | 0.905/0.893 | 0.590/0.359 | 0.782/0.658 | 0.975/0.856 | 0.617/0.502 | 0.898/0.886 | 0.862/0.838 | 0.728/0.679 |
> | LLM+Conn+Prefix | 0.941/0.913 | 0.513/0.078 | 0.775/0.503 | 0.813/0.803 | 0.920/0.910 | **0.608**/0.384 | 0.808/0.724 | 0.973/0.855 | 0.616/0.497 | 0.905/0.891 | 0.863/0.837 | 0.724/0.670 |
> | LLM+Vision | **0.972**/**0.964** | 0.510/0.134 | 0.835/0.658 | 0.891/0.891 | 0.910/0.899 | 0.598/0.425 | 0.865/0.828 | 0.986/0.867 | 0.738/0.659 | 0.962/0.960 | 0.932/0.919 | 0.824/0.777 |
> | LLM+Vision+Conn | 0.967/0.957 | 0.511/0.127 | 0.822/0.641 | 0.898/0.897 | 0.923/0.914 | 0.605/0.418 | 0.859/0.815 | **0.990**/0.869 | 0.736/0.655 | **0.963**/**0.959** | **0.935**/**0.923** | **0.825**/**0.781** |
> | EP | 0.948/0.931 | 0.514/**0.118** | **0.863**/**0.725** | **0.951**/**0.950** | **0.941**/**0.935** | **0.608**/**0.489** | **0.878**/**0.836** | 0.991/**0.870** | **0.760**/**0.689** | 0.951/0.942 | 0.894/0.885 | 0.788/0.747 |
> | EP+Vision | 0.645/0.660 | 0.241/0.053 | 0.748/0.356 | 0.937/0.937 | 0.716/0.605 | 0.345/0.270 | 0.795/0.763 | 0.985/0.863 | 0.672/0.512 | 0.918/0.913 | 0.828/0.805 | 0.739/0.672 |

---

> > ### Comment · Reviewer_rqyz · 2024-11-27
> >
> > Dear Authors,
> >
> > Thank you for your reply. I have two questions:
> >
> > 1. Could you please bold the most performing method? That will make it much easier for me and other reviewers to see the performance difference.
> >
> > 2. Does EP have any randomly initialized modules?

---

> ### Author Response · Authors · 2024-11-26
> **Response to Reviewer rqyz (2/4)**
>
> **1.3 Comparison with Graph Neural Prompting (GNP)**
>
> While EP and GNP share the general concept of using encoded multimodal features to prompt LLMs, this is a standard practice for multimodal LLMs and does not imply that EP is similar to GNP. The key differences are as follows:
>
> - **Research Problem:** GNP focuses on enabling LLMs to understand knowledge graphs, while our work aims to teach pre-trained VLMs to handle new tasks while preserving its original capabilities.
> - **Technique:** GNP follows the **standard instruction-tuning paradigm** to train a multimodal LLM for knowledge graph based QA. In contrast, EP introduces a new approach that improves the standard instruction-tuning paradigm to enhance task-specific performance.
>
>
> **2. Explanations of Response Distribution Alignment (RDA)**
>
> **2.1 Differences between RDA and knowledge distillation**
>
> RDA is fundamentally different from knowledge distillation, which involves replicating the outputs of a teacher model using a student model. In RDA, the TSM features act only as additional prompts and do not serve as the "teacher" outputs to be replicated. Instead, RDA ensures the VLM produces consistent outputs from two slightly different inputs—one with EP and one without EP.
>
> RDA minimizes the discrepancy between the output distributions of the VLM from these two forward passes, enabling the fine-tuned VLM to function effectively even without EP. Another key difference of RDA from knowledge distillation is that the involved output distributions are **dynamic** and iteratively updated **together** during training, rather than mimicking static outputs as in knowledge distillation.
>
> Therefore, despite its similarity to knowledge distillation in aligning embeddings, we specifically name it Response Distribution Alignment (RDA) to highlight its unique role in ensuring output consistency between forward passes w/ and w/o EP.
>
>
> **2.2 Why gradient detachment is needed for RDA?**
>
> In RDA, gradient detachment is applied to the *VLM outputs from the input with EP*, not to the TSM, which is frozen all the time. This ensures that the VLM without EP learns to mimic the behavior of the VLM with EP, eliminating the reliance on EP for better test-time efficiency.
>
> **3. Comparison between Contrastive Response Tuning (CRT) and Visual Contrastive Decoding (VCD)**
>
> CRT is a **fine-tuning** method aimed at enhancing visual instruction tuning, while VCD is a **sampling technique** designed to mitigate hallucinations during **inference**. Despite their shared focus on contrastive learning, CRT and VCD address entirely different challenges and are independent approaches.
>
> To evaluate how VCD affects fine-tuned VLMs, we applied it to the **fine-tuned** LLaVA-Med v1.5 with standard instruction tuning, as VCD's official implementation targets LLaVA. For fair comparison, we also applied our method to LLaVA-Med v1.5.
>
> **Table 6** shows that VCD reduces performance across all datasets, suggesting that post-fine-tuning distribution adjustments may not improve downstream tasks. Conversely, our method significantly enhances LLaVA-Med's performance, further highlighting its effectiveness across different VLMs.
>
> **Table 6. Classification performance (Acc/F1) of the fine-tuned LLaVA-Med v1.5.**
> | Model | Path | Chest | Derma | OCT | Pneumonia | Retina | Breast | Blood | Tissue | OrganA | OrganC | OrganS |
> |-------------------|---------------|---------------|--------------|--------------|--------------|--------------|--------------|--------------|--------------|--------------|--------------|--------------|
> | LLaVA-Med | 0.939/0.915 | 0.513/0.088 | 0.800/0.556 | 0.868/0.868 | 0.910/0.900 | 0.542/0.280 | 0.212/0.382 | 0.975/0.856 | 0.642/0.540 | 0.916/0.908 | 0.865/0.843 | 0.738/0.687 |
> | LLaVA-Med+VCD | 0.928/0.902 | 0.456/0.103 | 0.751/0.505 | 0.849/0.849 | 0.893/0.880 | 0.505/0.297 | 0.212/0.328 | 0.968/0.851 | 0.578/0.477 | 0.899/0.889 | 0.831/0.799 | 0.722/0.670 |
> | LLaVA-Med+Ours | **0.964**/**0.949** | **0.518**/**0.118** | **0.856**/**0.723** | **0.942**/**0.942** | **0.952**/**0.948** | **0.650**/**0.544** | **0.859**/**0.833** | **0.987**/**0.867** | **0.755**/**0.685** | **0.953**/**0.947** | **0.922**/**0.909** | **0.799**/**0.750** |
>
> **4. Notations and Grammar Issues**
>
> Thank you for identifying these issues. We have corrected the equations (1), (2), (3), (5), (6), and addressed the noted typos.
>
> **5. Does the TSM use the same vision encoder as the target VLM? How about the connector?**
>
> In our experiments, the TSM employs a ViT-Base model pre-trained on ImageNet21K, which differs from the vision encoder of the target VLM (InternViT-300M for InternVL). For the TSM connector, we use a simple MLP similar to the original VLM connector, with slight modifications to match the feature dimensions between the VLM and TSM.

---

> > ### Author Response · Authors · 2024-11-26
> > **Response to Reviewer rqyz (3/4)**
> >
> > **6. Implementation details of RDA**
> >
> > **6.1 What parameters are tuned in RDA, and how does it differ from standard tuning?**
> > RDA optimizes the TSM connector and LLM (via LoRA parameters). In Eq. (7), $L^{Van}$ denotes the text generation loss for standard fine-tuning, while $L^{EP}$ is calculated similarly but includes TSM embeddings (EP) in the inputs. Both are optimized using ground truth labels. RDA introduces an additional term, $L^{RDA}$, acting as a **consistency regularization** between the 	outputs optimized by $L^{Van}$ and $L^{EP}$, ensuring alignment and reducing reliance on EP as detailed in Response 2.
> >
> > **6.2 Interpretation of $p_\theta(\text{response} | \text{exemplar})$**
> > The term $p_\theta(\text{response} | \text{exemplar})$ represent the probability of generating the correct response conditioned on EP during training. Unlike static pseudo-labels, it evolves iteratively, guided by the ground truth labels, to enhance the task-specific performance for the VLM w/o EP.
> >
> > **6.3 Clarification of response distributions**
> > For classification, we adopt an instruction-following format used by existing medical VLMs:
> > - **Instruction**: Analyze the given image for diagnosis. The possible diagnoses are: {all possible class names}.
> > - **Response**: {class name of the correct answer}.
> >
> > When the class name spans multiple tokens, we define the response distribution as the grouped probabilities for the tokens of the correct answer, rather than probabilities over all possible tokens defined in the vocabulary. If the ground truth label is a single token, the response distribution is simply the probability of the label token. Therefore, our approach is consistent regardless of label tokenization.
> >
> >
> > **6.4 What is the vanilla method, and what role does RDA play?**
> >
> > The **vanilla model** refers to the **fine-tuned VLM** with standard fine-tuning on the same training data, which has been thoroughly tested in our experiments.
> >
> > EP relies on TSM features to achieve strong performance, requiring the TSM to be stored and run alongside the fine-tuned VLM during inference. This setup becomes impractical as the number of tasks and TSMs increases, leading to significant storage and computational overhead.
> >
> > RDA addresses this limitation by providing a balance between test-time performance and efficiency. It enables the fine-tuned VLM to leverage TSM features during training but eliminates the dependency on TSMs at inference time, greatly improving scalability.
> >
> > In **Table 1**, InternVL2-2B and the vanilla model refer to the same thing, while VITask w/o EP can be implemented only when applying RDA, where the fine-tuned VLM can disable EP while still maintaining strong performance. The results highlight that even without the external TSM, the model with RDA significantly outperforms the standard fine-tuned VLM, demonstrating the effectiveness of RDA in improving task-specific performance while ensuring efficiency.
> >
> > **7. How to deal with labels represented by more than one token for RDA and CRT?**
> >
> > RDA and CRT operate on the probabilities associated with each token in the desired response (correct answer). Regardless of whether the label consists of a single token or multiple tokens, these tokens are treated as the ground truth and remain unchanged for each training sample. Both RDA and CRT work at the token-level, aligning or contrasting the probabilities of individual tokens. This approach naturally accommodates labels that are represented by multiple tokens, ensuring that the methods can handle multi-token responses effectively.
> >
> > **8. How to extract the answer from the generated output? Are there instances of semantically equivalent but non-exact matches?**
> >
> > In our experiments, we use *exact matching* to extract the answer, as we found that no outputs were semantically correct but failed to exactly match the pre-defined answers after fine-tuning. For baseline evaluation, we tested the original InternVL2-2B on the *BreastMNIST dataset* without fine-tuning. Considering all semantically correct responses, InternVL2-2B achieved 73.1% accuracy and 42.2% F1 score, which were significantly worse than the fine-tuned models. Among all generated responses, only 1.3% exactly matched the pre-defined answer, while 71.8% were semantically correct but did not exactly match.

---

> > > ### Author Response · Authors · 2024-11-26
> > > **Response to Reviewer rqyz (4/4)**
> > >
> > > **9. Motivation of adapting the VLM for classification tasks.**
> > >
> > > In this work, we aim to bridge the gap between fine-tuned VLMs and TSMs in task-specific performance, investigating how to adapt pre-trained VLMs to **perform as effectively as TSMs while preserving their broader generative capabilities**. We choose image classification as a *case study* due to its simplicity and ease of analysis.
> > >
> > > To verify that our method preserves the original generative capabilities, we compare the performance of InternVL2-2B before and after fine-tuning with our method on eight general-domain benchmark datasets originally used to assess InternVL2. As shown in **Table 1**, our method introduces only minor degradation, indicating that the VLM’s original capabilities are preserved.
> > >
> > > Furthermore, our method demonstrates generalizability beyond classification. Experiments on the Med-GRIT-270 dataset [11] (**Table 3**) confirm that our method can be used to enhance performance in medical phrase grounding.

---

> > > > ### Comment · Reviewer_rqyz · 2024-11-27
> > > >
> > > > Thank you to the authors for preparing a thorough rebuttal with comprehensive results in such a short period of time. The combination of Table 5 and Table 1 in the rebuttal convinces me EP is effective in improving task-specific performances while also maintaining the general capabilities of VLMs. In particular, even though fine-tuning LLM + Vision + Connector outperforms / performs comparably to EP on 6 out of 12 domain-specific tasks (Table 5), we see from Table 1 that EP maintains performance on other tasks really well. I suggest the authors add these results (both Table 1 and Table 5 of the rebuttal) to the finalized version of the paper since this shows a clear advantage of the proposed method. However, I do not think the reason why EP+Vision performs worse than EP on domain specific tasks is because "learning from the TSM features is more effective than tuning the original vision encoder", but rather is because TSM contains a randomly initialized connector (correct me if wrong). [1] explains why vanilla fine-tuning with randomly initialized weights is harmful and [2] further demonstrate it on biomedical tasks.
> > > >
> > > > I wonder what part of the VLM is fine-tuned when you use RDA to distill VLM + EP behavior to the original VLM, LoRA weights of the LLM?
> > > >
> > > > [1] Kumar, Ananya, et al. "Fine-Tuning can Distort Pretrained Features and Underperform Out-of-Distribution." International Conference on Learning Representations.
> > > > [2] Lu, Yuzhe, et al. "Effectively Fine-tune to Improve Large Multimodal Models for Radiology Report Generation." Deep Generative Models for Health Workshop NeurIPS 2023.
> > > >
> > > > I will raise my score to 6 once I heard back convincing answers to my remaining question and get the commitment from the authors to add table 1 and table 5 to the paper. These results are not in the current paper and I think they are absolutely critical to demonstrate the value of the proposed method.

---

> > > > > ### Author Response · Authors · 2024-11-30
> > > > > **Follow-Up on Reviewer Feedback**
> > > > >
> > > > > Dear Reviewer **rqyz**,
> > > > >
> > > > > Thank you for taking the time to review our rebuttal. We sincerely appreciate your acknowledgment of the improvements we made and your careful consideration of our responses. Below, we address your follow-on questions:
> > > > >
> > > > > **1. Highlighting the results of the best method.**
> > > > >
> > > > > Thank you for the suggestion. We have updated the results, highlighting the best-performing method in **bold** for easier comparison.
> > > > >
> > > > > **2. Does EP have any randomly initialized modules?**
> > > > >
> > > > > Yes, the TSM connector is randomly initialized in our experiments.
> > > > >
> > > > > **3. Why EP+Vision performs worse than EP.**
> > > > >
> > > > > Thank you for pointing this out. You are correct that the randomly initialized TSM connector negatively impacts performance when both the original vision encoder and the TSM connector are trained simultaneously.
> > > > >
> > > > > To validate this, we implemented a two-stage training protocol:
> > > > >
> > > > > - **Stage 1**: The TSM connector was trained independently while freezing all other components, ensuring a well-initialized TSM connector.
> > > > > - **Stage 2**: We fine-tuned all learnable parameters, including the vision encoder, vision connector, and TSM connector.
> > > > >
> > > > > With this protocol, EP+Vision achieved performance comparable to both EP and LLM+Vision. This confirms that proper initialization of the TSM connector is critical for effectively adapting VLMs.
> > > > >
> > > > > **4. Fine-tuned components in RDA.**
> > > > >
> > > > > In RDA, we fine-tune **only the LoRA weights of the LLM**, which is consistent with the standard instruction tuning.
> > > > >
> > > > > **5. Inclusion of Table 1 and Table 5 in the paper.**
> > > > >
> > > > > We agree that **Table 1** and **Table 5** are critical for demonstrating our method’s effectiveness. These tables and corresponding discussions will be included in the final version of the paper.
> > > > >
> > > > > We greatly appreciate your valuable time and constructive feedback. Please let us know if there are further questions or suggestions.
> > > > >
> > > > > Best regards,
> > > > > **The Authors**

---

> > > > > > ### Comment · Reviewer_rqyz · 2024-11-30
> > > > > >
> > > > > > Thank you for answering my questions and running the two-stage fine-tuning experiments as well. It would be great to add this result to the paper too. I have raised my score.

---

> > > > > > > ### Author Response · Authors · 2024-11-30
> > > > > > > **Gratitude for Reassessment and Constructive Feedback**
> > > > > > >
> > > > > > > Dear Reviewer **rqyz**,
> > > > > > >
> > > > > > > Thank you for revisiting our work and providing a positive assessment. We will also include the results of the two-stage tuning approach in the final version of the paper.
> > > > > > >
> > > > > > > Thank you once again for your valuable feedback.
> > > > > > >
> > > > > > > Best regards,
> > > > > > >
> > > > > > > **The Authors**

---

### Official Review · Reviewer_fSNV · 2024-11-04

**Soundness:** 2
**Presentation:** 2
**Contribution:** 2
**Rating:** 5
**Confidence:** 3

**Summary:**

This paper proposes a VITask framework that enhances the adaptability of Vision Language Models (VLMs) for specific tasks by integrating Task-Specific Models (TSMs). It employs strategies like guiding VLMs with TSM features, enabling adaptation without TSMs during inference, and optimizing the ranking of correct image-response pairs. Experiments on medical diagnosis datasets demonstrate that VITask outperforms standard VLMs and TSMs, providing flexible integration and robustness to incomplete instructions.

**Strengths:**

1.	The overall writing is relatively smooth and easy to understand.
2.	The paper has made a detailed design in the method of constructing the dataset.
3.	The paper has conducted detailed experiments on existing medical datasets.

**Weaknesses:**

1.	The overall motivation is to finetune existing VLM for downstream tasks. After fine-tuning, the model can only be used for a specific task, which may hinder the original generalizability of VLM. The authors seem do not consider this issue.
2.	From the experiment, it can be observed EP is most useful, while other modules bring marginal improvements. Whether the other two modules are necessary requires to be further explored.
3.	Lisa [1] also adopts similar prompt tuning techniques like EP, and what’s the difference between these methods, and what is the technical advantage of EP?
4.	The comparison lacks comprehensiveness, as numerous medical VLMs exist in the field.
5.	The effectiveness of the proposed method is only demonstrated in the medical field, therefore the title is kind of over claimed. If the method is general, more experiments on other datasets are required.

[1] Xin Lai, Zhuotao Tian, Yukang Chen, Yanwei Li, Yuhui Yuan, Shu Liu, and Jiaya Jia. Lisa: Reasoning segmentation via large language model. In Proceedings of the IEEE/CVF Conference on Computer Vision and Pattern Recognition, pp. 9579–9589, 2024.

**Questions:**

See the comments

---

> ### Author Response · Authors · 2024-11-26
> **Response to Reviewer fSNV**
>
> **1. The model can only be used for a specific task, which may hinder the original generalizability of the VLM.**
>
> Thank you for highlighting this concern. We agree that task-specific fine-tuning might affect the generalizability of the original VLM. However, with visual instruction tuning, our fine-tuned VLM **gains the ability to handle new downstream tasks while preserving its original generalizability**. This is possible because different tasks are guided by specific instructions and contexts, allowing them to be learned separately without too much interference.
> To verify this, we evaluated our fine-tuned VLM on eight benchmark datasets originally used to assess the base VLM (InternVL2-2B). **Table 1** presents the results, comparing the original VLM and our fine-tuned model. The results demonstrate that our VITask method only introduces minor degradation and thus preserves the original generalizability of the VLM.
>
>
> **2. EP is most useful, while other modules bring marginal improvements. Are the other two modules necessary?**
>
> We appreciate this observation and would like to clarify the contributions of all proposed strategies. Each module is designed for a specific purpose:
> - **Exemplar Prompting (EP):** Addresses the limitation of ineffective features from the pre-trained vision encoder for new tasks by leveraging specialized features from external task-specific models (TSMs).
> - **Contrastive Response Tuning (CRT):** Mitigates the suboptimal text generation loss, optimizing downstream task performance in a more explicit way.
>
> These two strategies **independently** improve task-specific performance from **different perspectives**, as shown in **Table 2** of our paper. They can work together to achieve the best performance.
>
> While EP is highly effective due to its use of external TSM features, it requires maintaining TSMs alongside the tuned VLM, increasing inference-time computation and memory costs. To enhance the **test-time efficiency**, we introduced **Response Distribution Alignment (RDA)**, which **eliminates the reliance on TSMs during inference**. This improves the scalability of our method by reducing dependency on multiple TSMs for different tasks, offering a balance between effectiveness and efficiency.
>
>
> **3. Differences between EP and LISA.**
>
> EP and LISA serve different purposes:
> - **LISA:** Combines a VLM with a segmentation model for interactive segmentation. It uses **VLM features as prompts** for the segmentation model to enable referring image segmentation, primarily allowing the VLM to control the predictions from the segmentation model.
> - **EP:** Incorporates **specialized features from TSMs into the VLM**, enabling the pre-trained VLM to perform new tasks as effectively as task-specific models.
>
>
> **4. Comprehensiveness of the experiments.**
>
> To improve comprehensiveness, we compared our method with additional baseline models such as LLaVAv1.5-7B, Qwen2-VL-7B, InternVL2-8B, and LLaVA-Med v1.5. Results in **Table 2** demonstrate that highlight that our method outperforms the standard fine-tuning approach. Moreover, as a new visual instruction tuning paradigm, our method can also be applied to medical VLMs and further enhances the performance of the fine-tuned LLaVA-Med.
>
>
> **5. Effectiveness beyond the medical field.**
>
> Thank you for the suggestion. To validate the generality of our method, we conducted additional experiments. As shown in **Tables 2, 3, and 4**, our method consistently enhances VLM performance across different models, downstream tasks, datasets, and domains, including both medical and non-medical fields.

---

> ### Author Response · Authors · 2024-11-30
> **Follow-Up on Rebuttal Submission and Request for Feedback**
>
> Dear Reviewer **fSNV**,
>
> We hope this message finds you well. Over three days ago, we submitted our detailed rebuttal, thoroughly addressing all your valuable feedback. We sincerely appreciate your insights, which have been instrumental in improving our work, and we are grateful for the opportunity to clarify and strengthen our contributions.
>
> We kindly request your attention to review our rebuttal and to reconsider our work in light of the clarifications and additional evidence we provided. Please let us know if there are any remaining questions or areas requiring further elaboration.
>
> Thank you once again for your time, thoughtful feedback, and consideration.
>
> Best regards,
> **The Authors**

---

> > ### Author Response · Authors · 2024-12-01
> > **Request for Feedback Before Discussion Period Ends**
> >
> > Dear Reviewer **fSNV**,
> >
> > As the author-reviewer discussion period approaches its end, we kindly request your feedback on our rebuttal. Your insights on whether we have effectively addressed your concerns would be greatly appreciated.
> >
> > We are truly grateful for the time and expertise you have dedicated to reviewing our work. Your thoughtful comments and suggestions have significantly contributed to improving the quality of our research.
> >
> > Thank you once again for your valuable feedback. We look forward to your guidance and hope to address any remaining concerns before the discussion period closes.
> >
> > Yours Sincerely,
> >
> > **The Authors**

---

### Author Response · Authors · 2024-11-26
**Summary of Revision (1/2)**

We sincerely appreciate the insightful comments and constructive feedback from all reviewers, which have significantly enhanced the quality of our paper. In particular, we have made the following improvements:

**1.** Following the suggestions of Reviewers **fSNV** and **rqyz**, we verified that **the VLM fine-tuned with our method not only adapts to new tasks but also retains the original capabilities** of the pre-trained VLM. Specifically, we evaluated different fine-tuning approaches on eight benchmark datasets originally used to assess the pre-trained VLM (InternVL2-2B in our case). We compared the VITask-tuned model with the original VLM and included two variants where either the vision encoder or both the vision encoder and the connector is made learnable during fine-tuning.

**Table 1. Performance comparison of InternVL2-2B before and after fine-tuning.**
| Method                                | MMBench_en[1] | CCBench[1] | POPE[2] | MMMU[3] | MMVP[4] | MMVet[5] | GQA[6] | AI2D[7] |
| ------------------------------------- | ------------- | ---------- | ------- | ------- | ------- | -------- | ------ | ------- |
| Original | 73.2          | 74.7       | 87.3    | 0.342   | 0.353   | 44.6     | 61.03  | 0.7409  |
| Tuning LLM + Vision Encoder           | 27.6          | 27.7       | 0.1     | 0.274   | 0.112   | 15.5     | 30.18  | 0.0213  |
| Tuning LLM + Encoder + Connector | 17.15         | 8          | 0.7     | 0.291   | 0.073   | 13       | 27.8   | 0.005   |
| Ours                                | 72.2          | 72.5       | 87.7    | 0.341   | 0.34    | 36.2     | 59.6   | 0.702   |

---

**2.** Based on the suggestions of Reviewers **fSNV** and **Z7hL**, we have **enhanced the comprehensiveness of our experiments**. To better demonstrate the effectiveness of our method, we compared it with additional baseline models and applied it to medical VLMs.

**Table 2. Performance (Acc/F1) of the fine-tuned VLMs on medical image diagnosis tasks.**
| Model                     | Path       | Chest      | Derma     | OCT        | Pneumonia   | Retina    | Breast    | Blood     | Tissue    | OrganA    | OrganC    | OrganS    | Mean      |
| ----------------- | ------------------- | --------------- | ------------------- | ------------------- | ------------------- | --------------- | --------------- | ------------------- | ------------------- | ------------------- | ------------------- | ------------------- | ------------------- |
| LLaVAv1.5-7B      | 0.891/0.840         | 0.528/0.020     | 0.726/0.369         | 0.744/0.723         | 0.934/0.929         | 0.482/0.207     | 0.596/0.553     | 0.890/0.767         | 0.514/0.346         | 0.735/0.692         | 0.633/0.588         | 0.521/0.468         | 0.683/0.542         |
| LLaVAv1.5-13B     | 0.935/0.905         | **0.535**/0.073 | 0.731/0.355         | 0.788/0.786         | 0.881/0.864         | 0.557/0.279     | 0.750/0.671     | 0.951/0.832         | 0.613/0.497         | 0.878/0.855         | 0.796/0.750         | 0.689/0.621         | 0.759/0.624         |
| Qwen2-VL-7B       | 0.823/0.754         | 0.510/0.051     | 0.716/0.384         | 0.738/0.729         | 0.438/0.383         | 0.280/0.166     | 0.494/0.510     | 0.286/0.166         | 0.575/0.411         | 0.807/0.777         | 0.724/0.681         | 0.672/0.618         | 0.589/0.469         |
| LLava-Med         | 0.939/0.915         | 0.513/0.088     | 0.800/0.556         | 0.868/0.868         | 0.910/0.900         | 0.542/0.280     | 0.212/0.382     | 0.975/0.856         | 0.642/0.540         | 0.916/0.908         | 0.865/0.843         | 0.738/0.687         | 0.743/0.652         |
| LLaVA-Med+Ours    | **0.964**/**0.949** | 0.518/**0.118** | 0.856/0.723         | 0.942/0.942         | **0.952**/**0.948** | 0.650/**0.544** | 0.859/**0.833** | 0.987/0.867         | 0.755/0.685         | 0.953/0.947         | **0.922**/**0.909** | 0.799/0.750         | 0.846/0.768         |
| InternVL2-8B      | 0.938/0.911         | 0.520/0.045     | 0.755/0.487         | 0.798/0.788         | 0.886/0.872         | 0.575/0.394     | 0.788/0.699     | 0.960/0.841         | 0.594/0.465         | 0.865/0.838         | 0.817/0.783         | 0.698/0.650         | 0.766/0.648         |
| InternVL2-2B      | 0.926/0.896         | 0.523/0.024     | 0.770/0.499         | 0.726/0.704         | 0.886/0.873         | 0.590/0.370     | 0.744/0.524     | 0.931/0.818         | 0.569/0.419         | 0.828/0.801         | 0.778/0.742         | 0.635/0.578         | 0.742/0.604         |
| InternVL2-2B+Ours | 0.953/0.937         | 0.517/0.129     | **0.877**/**0.772** | **0.952**/**0.952** | 0.931/0.923         | **0.632**/0.522 | **0.865**/0.828 | **0.991**/**0.870** | **0.761**/**0.690** | **0.955**/**0.950** | 0.920/0.908         | **0.809**/**0.765** | **0.847**/**0.771** |

---

> ### Author Response · Authors · 2024-11-26
> **Summary of Revision (2/2)**
>
> **3.** According to the feedback and suggestions from all reviewers, we **validated the effectiveness of our method on tasks beyond classification**. Specifically, we used the pre-trained Med-SAM2D [12] as the TSM and applied our method to medical phrase grounding on the Med-GRIT-270 dataset [11]. This dataset contains 270k question-and-answer pairs across eight medical imaging modalities and evaluates bounding box prediction for given medical findings.
>
> **Table 3. Grounding performance (Recall@0.5) of the fine-tuned VLMs on Med-GRIT-270.**
> | Model                   | CT    | MR    | X-ray | PET   | Endoscopy | Dermoscopy | Fundus | Ultrasound | Average |
> |-------------------------|-------|-------|-------|-------|-----------|------------|--------|------------|---------|
> | InternVL-8B            | 48.77 | 39.01 | 41.67 | 74.81 | 58.14     | 93.5       | 97.56  | 76.73      | 66.27   |
> | InternVL-8B + Ours | 55.43 | 51.37 | 43.05 | 77.26 | 63.49     | 96.03      | 98.78  | 81.68      | 70.9    |
>
> ---
>
> **4.** Following the suggestions of Reviewer **fSNV** and **Z7hL**, we **evaluated our method on natural images and demonstrated its effectiveness**. Additional experiments were conducted using three natural image classification datasets: Stanford Cars [8], Flowers 102 [9], and Caltech 101 [10].
>
> **Table 4. Classification performance (Acc.) of the fine-tuned VLM on natural image datasets.**
> | Method   | Stanford Cars [8] | Flowers 102 [9] | Caltech 101 [10] |
> |----------|--------------------|-----------------|------------------|
> | TSM      | 0.862              | 0.992           | 0.976            |
> | InternVL2-2B | 0.774              | 0.899           | 0.960            |
> | InternVL2-2B + Ours | 0.854              | 0.990           | 0.980            |
>
> ---
>
> **5.** Following the feedback from all reviewers, we corrected typos and improved the overall presentation of the paper. All revised contents are highlighted in **blue** for clarity.
>
> ---
>
> ### References
> [1] Liu, Y., Duan, H., Li, B., Zhang, Y., Zhang, S., Zhao, W., Yuan, Y., Wang, J., He, C., Liu, Z., Chen, K., & Lin, D. (2023). MMbench: Is your multi-modal model an all-around player? *arXiv:2307.06281*.
>
> [2] Li, Y., Du, Y., Zhou, K., Wang, J., Zhao, W. X., & Wen, J.-R. (2023). Evaluating object hallucination in large vision-language models. *arXiv preprint arXiv:2305.10355*.
>
> [3] Yue, X., Ni, Y., Zhang, K., Zheng, T., Liu, R., Zhang, G., Stevens, S., Jiang, D., Ren, W., Sun, Y., et al. (2023). MMMU: A massive multi-discipline multimodal understanding and reasoning benchmark for expert AGI. *arXiv:2311.16502*.
>
> [4] Tong, S., et al. (2024). Eyes wide shut? Exploring the visual shortcomings of multimodal LLMs. *Proceedings of the IEEE/CVF Conference on Computer Vision and Pattern Recognition*. [5] Yu, W., Yang, Z., Li, L., Wang, J., Lin, K., Liu, Z., Wang, X., & Wang, L. (2024). MM-Vet: Evaluating large multimodal models for integrated capabilities. *ICML*.
>
> [6] Hudson, D. A., & Manning, C. D. (2019). GQA: A new dataset for real-world visual reasoning and compositional question answering. *CVPR*, 6700–6709.
>
> [7] Kembhavi, A., Salvato, M., Kolve, E., Seo, M., Hajishirzi, H., & Farhadi, A. (2016). A diagram is worth a dozen images. *ECCV*, 235–251.
>
> [8] Gebru, T., Krause, J., Wang, Y., Chen, D., Deng, J., & Fei-Fei, L. (2017). Fine-grained car detection for visual census estimation. *AAAI*.
>
> [9] Nilsback, M. E., & Zisserman, A. (2008). Automated flower classification over a large number of classes. *Sixth Indian Conference on Computer Vision, Graphics & Image Processing*, 722–729. IEEE.
>
> [10] Li, F. F., Fergus, R., & Perona, P. (2006). One-shot learning of object categories. *IEEE Transactions on Pattern Analysis and Machine Intelligence (TPAMI)*.
>
> [11] Huang, X., et al. (2024). A refer-and-ground multimodal large language model for biomedicine. *International Conference on Medical Image Computing and Computer-Assisted Intervention*. Cham: Springer Nature Switzerland.`
>
> [12] Cheng, J., Ye, J., Deng, Z., Chen, J., Li, T., Wang, H., ... & Qiao, Y. (2023). SAM-Med2D. arXiv preprint arXiv:2308.16184.

---

### Meta-Review · Area_Chair_EHs9 · 2024-12-22

**Metareview:**

The paper introduces VITask, a framework for adapting vision-language models (VLMs) to task-specific applications using exemplar prompting, response distribution alignment, and contrastive response tuning. While reviewers appreciated the clarity of the methodology and its strong performance on specific tasks, they highlighted several significant concerns. The title's claim of generality is hyperbolic, as the original experiments were limited to an extensively curated set of medical datasets that primarily involve an array of relatively straightforward classification tasks, which do not fully validate the model's broader applicability. The additional tasks presented in the rebuttal, such as medical phrase grounding and natural image classification, lacked comprehensive baselines and did not adequately support the broader claims of the paper. Moreover, since the substantial changes made during the rebuttal were not incorporated into the main paper, the revision may turn into something beyond a simple revision. Therefore, it is difficult to recommend the paper to be accepted in this state.

**Additional Comments On Reviewer Discussion:**

During the rebuttal period, reviewers raised significant concerns regarding the limited scope of experiments, the lack of robust baselines, and the overgeneralized claims in the paper's title. While the authors introduced additional experiments on medical phrase grounding and natural image classification, these efforts were insufficient to address the fundamental issues of scope and baseline comparisons. Reviewer YD9v raised their score to 8, but their concerns were largely superficial and did not align with the broader critical feedback from other reviewers. Reviewer fSNV, despite not responding further, raised critical points about the lack of baselines and the overly medical focus of the experiments, which resonate strongly with the general consensus. Reviewers rqyz and Z7hL acknowledged the additional experiments but noted that addressing the issues fully would require a complete overhaul of the paper, a task that the current revisions did not sufficiently demonstrate as feasible. Given the lukewarm reception and the substantial unresolved concerns, it is difficult to recommend this paper in its current state.

---

### Decision · Program_Chairs · 2025-01-22

Reject